# Effect of gut microbiota on depressive-like behaviors in mice is mediated by the endocannabinoid system

Grégoire Chevalier [1], Eleni Siopi [2,7], Laure Guenin-Macé[3,7], Maud Pascal[1,2,7], Thomas Laval[3], Aline Rifflet[4], Ivo Gomperts Boneca [4], Caroline Demangel [3], Benoit Colsch [5], Alain Pruvost [5], Emeline Chu-Van[5], Aurélie Messager[5], François Leulier [6], Gabriel Lepousez [2,8], Gérard Eberl [1,8✉] & Pierre-Marie Lledo [2,8✉]

Depression is the leading cause of disability worldwide. Recent observations have revealed an association between mood disorders and alterations of the intestinal microbiota. Here, using unpredictable chronic mild stress (UCMS) as a mouse model of depression, we show that UCMS mice display phenotypic alterations, which could be transferred from UCMS donors to naïve recipient mice by fecal microbiota transplantation. The cellular and behavioral alterations observed in recipient mice were accompanied by a decrease in the endocannabinoid (eCB) signaling due to lower peripheral levels of fatty acid precursors of eCB ligands. The adverse effects of UCMS-transferred microbiota were alleviated by selectively enhancing the central eCB or by complementation with a strain of the *Lactobacilli* genus. Our findings provide a mechanistic scenario for how chronic stress, diet and gut microbiota generate a pathological feed-forward loop that contributes to despair behavior via the central eCB system.

[1] Microenvironment and Immunity Unit, INSERM U1224, Institut Pasteur, Paris, France. [2] Perception and Memory Unit, CNRS UMR3571, Institut Pasteur, Paris, France. [3] Immunobiology of Infection Unit, INSERM U1221, Institut Pasteur, Paris, France. [4] Biology and Genetics of Bacterial Cell Wall Unit, CNRS UMR2001, INSERM, Equipe Avenir, Institut Pasteur, Paris, France. [5] Université Paris Saclay, CEA, INRAE, Médicaments et Technologie pour la Santé (MTS), Gif-sur-Yvette, France. [6] Institut de Génomique Fonctionnelle de Lyon, Université de Lyon, Ecole Normale Supérieure de Lyon, CNRS UMR 5242, Lyon, France. [7] These authors contributed equally: Eleni Siopi, Laure Guenin-Macé, Maud Pascal. [8] These authors jointly supervised this work: Gabriel Lepousez, Gérard Eberl, Pierre-Marie Lledo. ✉email: gerard.eberl@pasteur.fr; pierre-marie.lledo@pasteur.fr

D epression is the leading cause of disability worldwide, currently affecting >300 million people[1]. Despite the prevalence of depression and its considerable economic impact, its pathophysiology remains highly debated. Yet, a better understanding of the mechanisms leading to depression is a prerequisite for developing efficient therapeutic strategies. However, unraveling the pathophysiology of depression is challenging, as depressive syndromes are heterogeneous and their etiologies likely to be diverse. Experimental and genetic studies have yielded several mechanisms, including maladaptive responses to stress with hypothalamic–pituitary–adrenal (HPA) axis dysregulation, inflammation, reduced neuroplasticity, circuit dysfunctions, and perturbation in neuromodulatory systems, such as monoaminergic and endocannabinoid (eCB) systems.

A number of studies converge to indicate hippocampal alterations as critical in the pathogenesis of depression. For instance, hippocampal volume loss is a hallmark of clinical depression[2]. Likewise, rodent studies have demonstrated that chronic stress-induced depression impair adult hippocampal neurogenesis[3]. Furthermore, impaired hippocampal neurogenesis results in depressive-like behaviors in rodent, in part because hippocampal neurogenesis buffers the over-reactivity of the HPA axis in response to stress[4,5]. In that line, antidepressants and alternative antidepressant interventions stimulate adult hippocampal neurogenesis, which in turn dampens stress responses and restores normal behavior[6,7]. Adult hippocampal neurogenesis is thus considered as an important causal factor and a key marker of depression, although a direct causal link is still missing in human depression[8].

Over the past decade, the impact of the symbiotic microbiota on numerous host functions has been increasingly recognized. The wide variety of intestinal microbes affects many processes, including immunity[9], metabolism[10], and the central nervous system[11]. In depressed patients, alterations in the composition of the intestinal microbiota (named dysbiosis) have been characterized[12,13]. Furthermore, numerous studies on animal models have shown that the microbiota modulates anxiety[14] and onset of neurological diseases associated with circuit dysfunctions[15] by releasing bacterial metabolites that can directly or indirectly affect brain homeostasis. In that line of ideas, microbiota from depressed patients alter behavior when transferred to antibiotic-treated rats[16] and murine gut microbiota dysbiosis is associated with several neurobiological features of depression, such as low-grade chronic inflammation[17], abnormal activity of the HPA axis[18], and decreased adult neurogenesis[3]. The notion that microbiota is a critical node in the gut–brain axis is also supported by the observation that colitis, which depends on the gut microbiota, shows significant comorbidities with depression[19]. Finally, probiotic intervention has been shown to influence emotional behavior in animal models of depression[20] and improve mood in depressive patients[21]. However, the molecular mechanisms linking intestinal microbiota and mood disorders remain largely unknown, partly due to the lack of experimental models.

To explore a causative role of the gut microbiota in stress-induced depressive behaviors, we used unpredictable chronic mild stress (UCMS), a mouse model of depression, and fecal microbiota transfer (FMT) from stressed donors to naive mice. We found that the microbiota transplantation transmits the depressive behavioral symptoms and reduces adult neurogenesis of the recipient mice. Metabolomic analysis reveals that recipient mice developed an altered fatty acid metabolism characterized by deficits in lipid precursors for eCBs, which resulted in impaired activity of the eCB system in the brain. Increase of the eCB levels after pharmacological blocking of the eCB degrading enzymes, or complementation of the diet with arachidonic acid (AA), a precursor of eCBs, is sufficient to alleviate both the microbiota-induced depressive-like behaviors and hippocampal neurogenesis impairments in recipient mice. Lastly, our study reveals that UCMS induced a gut microbiota dysbiosis characterized by a decrease in Lactobacilli abundance also observed in recipient mice. Complementation of UCMS-recipient mice with a strain of the Lactobacilli genus is sufficient to increase both eCB brain levels and hippocampal neurogenesis, alleviating the microbiota-induced despair behavior.

## Results

**Microbiota influences depressive-like behaviors and neurogenesis.** To establish a depressive-like state in mice, we submitted C57BL/6J mice for 8 weeks to UCMS, a well-defined mouse model of stress-induced depression[22] (Fig. 1a and Supplementary Table S1). Consistent with previous reports, UCMS mice developed depressive-like behaviors, as shown by increased feeding latency in the novelty suppressed feeding test as compared to control mice (Fig. 1b), even though feeding drive was not affected (Supplementary Fig. 1A). This behavior reflects both anxiety and anhedonia. However, UCMS mice did not develop increased anxiety, as determined by the light/dark box (LDB) test (Fig. 1c). Furthermore, UCMS mice showed increased grooming latency (Fig. 1d) and decreased self-grooming behavior in the splash test (Supplementary Fig. 1B, C), reflecting symptoms of depression such as apathetic behavior. The depressive-like state seen in UCMS mice was further confirmed in two prototypical tests for assessing depressive-like behaviors, the tail suspension test and the forced swim test (also named behavioral despair tests). UCMS mice showed increased immobility time in these two tests compared to control mice (Fig. 1e, f). We also observed that UCMS mice gained significantly less weight over time than control mice, as previously reported[23] (Supplementary Fig. 1D). Altogether, these different behavioral tests demonstrate that 8 weeks of UCMS induce depressive-like but not anxiety-like behaviors in C57BL/6 mice.

As the reduction of adult hippocampal neurogenesis is a hallmark of depression, we tested whether UCMS affected the number of adult-born neurons in the dentate gyrus (DG) of the hippocampus. The decreased number of proliferating neural stem cells labeled with the cell proliferation marker Ki67 (Fig. 1g, h), and of doublecortin (DCX)$^+$ cells, a marker for newborn immature neurons (Fig. 1g, i), shows that UCMS mice exhibit reduced hippocampal neurogenesis.

We next assessed whether the transplantation of gut microbiota from UCMS mice to naive unstressed hosts was sufficient to transfer the hallmarks of depressive-like state. To this end, we transferred the fecal microbiota of control or stressed mice to adult germ-free mice (Fig. 1a). Eight weeks after FMT, recipients of UCMS microbiota showed depressive-like behaviors in both the tail suspension and the forced swim tests (Fig. 1e, f), which were confirmed in the splash test (Fig. 1d and Supplementary Fig. 1F, G) and the novelty suppressed feeding test (Fig. 1b and Supplementary Fig. 1E). As in UCMS donors, recipient mice did not express anxiety-related behaviors (Fig. 1c). Similar results were obtained when UCMS microbiota was transferred to recipient specific-pathogen free (SPF) mice that were treated with broad-spectrum antibiotics for 6 days until 1 day prior to FMT (Supplementary Fig. 2). Because germ-free mice might exhibit some behavioral abnormalities due to sustained disruption in the microbiota–gut–brain axis, all subsequent experiments were performed using short-term antibiotic-treated recipient mice. Finally, recipients of UCMS microbiota also showed decreased proliferation of neural stem cells (Fig. 1g, h) and decreased production of new neurons in the hippocampus

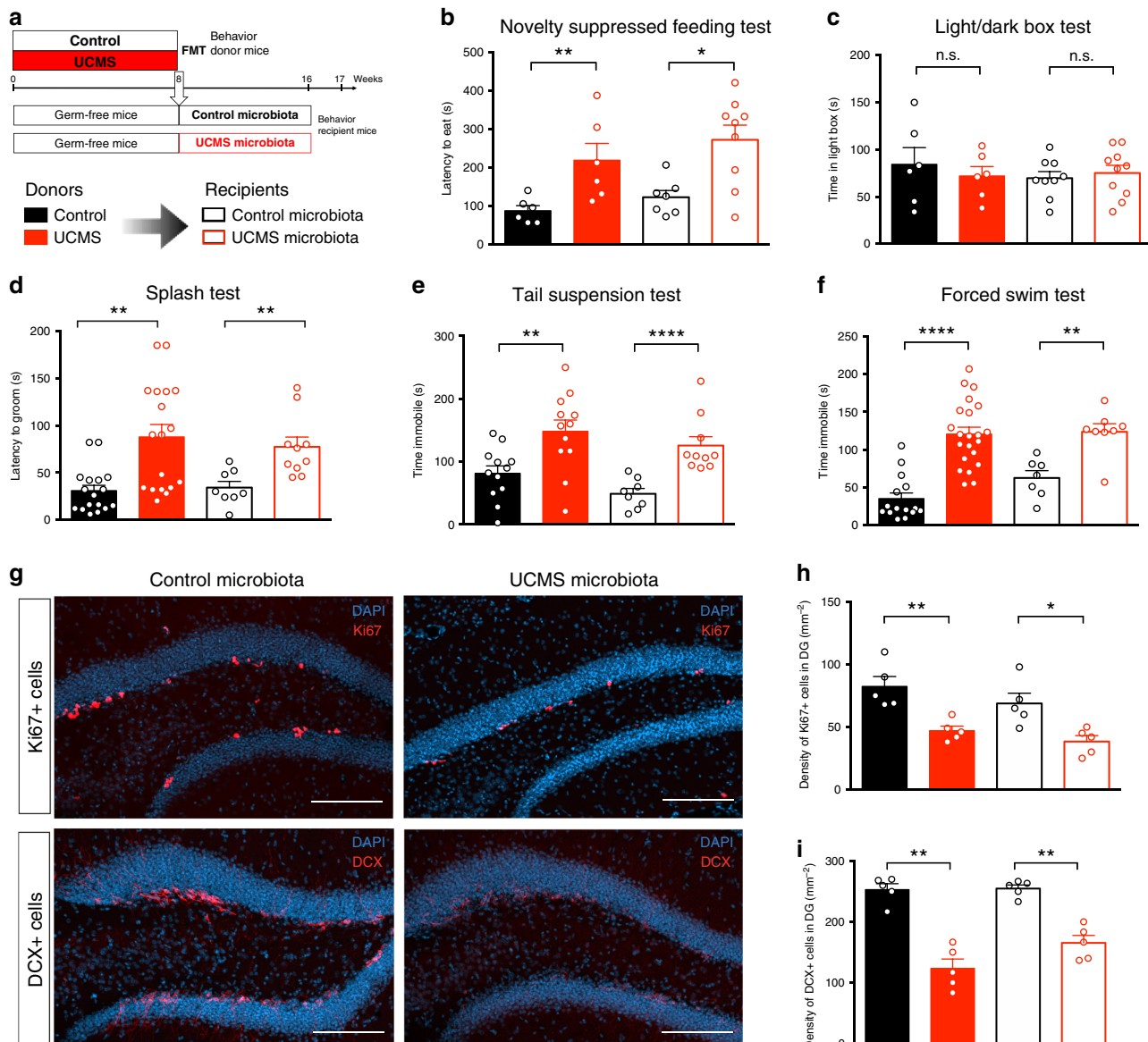

**Fig. 1 Microbiota from UCMS mice transfers depressive-like behaviors and reduces adult hippocampal neurogenesis. a** Experimental timeline of fecal microbiota transplantation (FMT) from control and UCMS mice, respectively, "Control microbiota" and "UCMS microbiota," to germ-free recipient mice. **b–f** Control mice (black bars), or mice subjected to UCMS (red bars), and mice recipient of the microbiota from Control (open black bars) or UCMS mice (open red bars), underwent different behavioral tests. **b** Latency to eat in a novel environment in the novelty suppressed feeding test for Control mice ($n =$ 6), UCMS mice ($n = 6$), Control microbiota-recipient mice ($n = 7$), and UCMS microbiota-recipient mice ($n = 9$). (Control vs UCMS, $P = 0.0087$; Control microbiota- vs UCMS microbiota-recipient mice, $P = 0.0229$); **c** Time spent in the light box in the light/dark box test for Control mice ($n = 6$), UCMS mice ($n = 6$), Control microbiota-recipient mice ($n = 9$), and UCMS microbiota-recipient mice ($n = 10$). (Control vs UCMS, $P = 0.6991$; Control microbiota- vs UCMS microbiota-recipient mice, $P = 0.6038$); **d** Latency to groom in the Splash test for Control mice ($n = 17$), UCMS mice ($n = 18$), Control microbiota-recipient mice ($n = 8$), and UCMS microbiota-recipient mice ($n = 10$). (Control vs UCMS, $P = 0.0004$; Control microbiota- vs UCMS microbiota-recipient mice, $P = 0.0012$; **e** Time spent immobile in the tail suspension test for Control mice ($n = 10$), UCMS mice ($n = 10$), Control microbiota-recipient mice ($n = 8$), and UCMS microbiota-recipient mice ($n = 10$). (Control vs UCMS, $P = 0.0043$; Control microbiota- vs UCMS microbiota-recipient mice, $P < 0.0001$; **f** Time spent immobile in the forced swim test for Control mice ($n = 15$), UCMS mice ($n = 22$), Control microbiota-recipient mice ($n = 7$), and UCMS microbiota-recipient mice ($n = 8$). (Control vs UCMS, $P < 0.0001$; Control microbiota- vs UCMS microbiota-recipient mice, $P = 0.0037$). **g** Representative images of Ki67 staining (top) and DCX staining (bottom) in the DG of the hippocampus, counterstained with DAPI (blue), in Control microbiota-recipient mice (left) and UCMS microbiota-recipient mice (right). **h** Quantitative evaluation of the density of Ki67$^+$ cells in the dentate gyrus (DG) of the hippocampus for Control mice ($n = 5$), UCMS mice ($n = 5$), Control microbiota-recipient mice ($n = 5$), and UCMS microbiota-recipient mice ($n = 5$). (Control vs UCMS, $P = 0.0079$; Control microbiota- vs UCMS microbiota-recipient mice, $P = 0.0159$). Scale bar: 100 μm. **i** Quantitative evaluation of the density of DCX$^+$ cells in the DG of the hippocampus for Control mice ($n = 5$), UCMS mice ($n = 5$), Control microbiota-recipient mice ($n = 5$), and UCMS microbiota-recipient mice ($n = 5$). (Control vs UCMS, $P = 0.0079$; Control microbiota- vs UCMS microbiota-recipient mice, $P = 0.0079$). For **b–i**, data are represented as mean ± s.e.m. Statistical significance was calculated using Mann–Whitney test ($^*P < 0.05$, $^{**}P < 0.01$, $^{****}P < 0.0001$, two tailed). Source data are provided as a Source data file.

(Fig. 1g, i). These data demonstrate that the hallmarks of depressive-like behaviors are transferable to naive recipient mice by the transplantation of fecal microbiota obtained from stressed-induced depressive mice.

**Microbiota alters fatty acid metabolism and eCB system**. We explored the possibility that UCMS microbiota triggered depressive-like behaviors through alterations of the host's metabolism. Metabolomic profiling of serum revealed a significant decrease in the levels of monoacylglycerols (MAG) and diacylglycerols (DAG) in both UCMS mice and recipients of UCMS microbiota, as compared to control and recipients of control microbiota (Fig. 2a). Furthermore, the n-6 polyunsaturated fatty acid (PUFA), AA (20:4n-6), its precursor linoleic acid (18:2n-6), and n6-PUFA biosynthesis intermediates were significantly decreased in the serum of both UCMS donors and recipients (Fig. 2b, c). This lipid loss was specific to short-chain fatty acids since levels of several medium- and long-chain fatty acylcarnitines rather increased in UCMS microbiota recipients (Supplementary Fig. 3A). This observation in the serum was also assessed in the hippocampus. For this, we performed a metabolomic analysis for PUFAs from lipidic extracts of hippocampus of recipient mice and observed a general tendency toward a decrease of both n-3 and n-6 PUFA in the hippocampus of UCMS microbiota-recipient mice (Supplementary Fig. 3E). Importantly, we did not observe any significant differences in kynurenine plasmatic concentration (Supplementary Fig. 4L) nor in baseline corticosterone level (Supplementary Fig. 4N) in recipient mice. Lastly, we also investigated the status of the gut immune system and observed no differences in both innate and adaptive immune cell populations (Supplementary Fig. 4A–K).

Such changes in the levels of fatty acids could originate from altered gut permeability and/or dysbiosis-induced lipid metabolism changes. To test the first hypothesis, we quantified fluorescence level in the serum following gavage with fluorescein isothiocyanate (FITC)–dextran and found no change in gut permeability (Supplementary Fig. 3F). To address the second hypothesis, we scrutinize several fatty acid metabolites and found that two precursors for the production of eCB, AA-containing DAG and n-6 PUFA, were dramatically reduced in recipient mice transplanted with UCMS microbiota but not with control microbiota. Interestingly, dysregulation of the eCB system and its main central receptor CB1 has been associated with the pathophysiology of depression both in humans and in UCMS model of depression[24,25].

Since previous studies have shown that activation of the CB1 receptors produces anxiolytic- and antidepressant-like effects, notably via the modulation of hippocampal neurogenesis[26–28], we investigated into more details the brain eCB system. We examined both the hippocampal production of eCB ligands and the activation level of the CB1 receptor pathway. As the AA-containing DAG and n-6 PUFA are precursors of the eCB 2-arachidonoylglycerol (2-AG), we first compared the levels of 2-AG (including 1(3)-AG, a product of chemical isomerization of 2-AG) in the hippocampus and serum of donor and recipient mice. Levels of hippocampal 2-AG, determined by mass spectrometry, revealed a significant decrease in both UCMS donors and recipients (Fig. 2d), with a strong inverse correlation found between the serum levels of 1-AG and the depressive state (Fig. 2e). Importantly, we did not observe any significant decrease of the other major eCB anandamide (AEA) in the hippocampus of UCMS microbiota recipients (Fig. 2f).

In the hippocampus, activation of CB1 receptors triggers mammalian target of rapamycin (mTOR) signaling. To evaluate whether deficiency in 2-AG leads to altered activity of the mTOR

pathway, we quantified phosphorylated (active) mTOR and its downstream effectors in both UCMS donor and recipient mice. mTOR phosphorylates the 70-kDa ribosomal protein S6 kinase (p70S6K) at T389 and the activated p70S6K in turn phosphorylates the ribosomal protein S6 (rpS6) at S235/236, which initiates mRNA translation. Donors and recipients of UCMS microbiota showed significantly decreased phosphorylation of mTOR (p-mTOR), p70S6K (p-p70S6K), and rpS6 (p-rpS6) (Fig. 2g–i). Collectively, these results demonstrate that the signature in lipid metabolism of UCMS microbiota comprises a deficiency in serum 2-AG precursors, lower content in hippocampal 2-AG, and breakdown of the mTOR signaling. Remarkably, these features were found to be transmittable to naive recipient mice following FMT.

**Restoration of eCB signaling normalizes behavior and neurogenesis**. To further demonstrate the role of defective eCB signaling in the depressive-like behaviors of mice transferred with UCMS microbiota, we next assessed whether enhancing eCB signaling, using pharmacological blockade of the 2-AG-degrading enzyme monoacylglycerol lipase (MAGL), could alleviate these phenotypes. Recipients of UCMS microbiota were treated with the MAGL inhibitor JZL184, or JZL184 together with rimonabant, a selective antagonist of CB1, every 2 days for 4 weeks starting 4 weeks after FMT (Fig. 3a). First, we confirmed that recipients of UCMS microbiota treated with JZL184 showed a significant increase in hippocampal levels of p-mTOR, p-p70S6K, and p-rpS6 as compared to vehicle-treated recipient mice of UCMS microbiota (Fig. 3b, c). Furthermore, consistent with enhanced eCB signaling, we confirmed that JZL184 enhanced the levels of 2-AG in the hippocampus (Fig. 3d). The effect of JZL184 was strictly CB1 dependent as it was reversed by the selective CB1 receptor antagonist rimonabant. As a consequence, JZL184 reduced depressive symptoms in recipients of UCMS microbiota, an effect that was blocked by rimonabant (Fig. 3e–h). To assess the relative contribution of central vs peripheral CB1 receptors in these depressive-like behaviors, we compared the effects of rimonabant to the effects of AM6545, a CB1 antagonist with limited brain penetrance. In contrast to rimonabant, AM6545 did not reverse the antidepressant effect of JZL184 (Fig. 3g, h), indicating that central CB1 signaling is necessary to alleviate depressive-like behaviors, at least in our model.

JZL184 also alleviates the detrimental effects of UCMS microbiota on adult hippocampal neurogenesis. JZL184 treatment rescued the proliferation and differentiation of neural stem cells in the hippocampus of UCMS microbiota recipients, an effect that was blocked by rimonabant (Fig. 4a–d). The survival of newly generated neurons was also increased in the hippocampus of mice treated with JZL184, and blocked by rimonabant, as shown by the quantification of newborn neurons labeled with the DNA synthesis marker 5-ethynyl-2'-deoxyuridine (EdU) administered 4 weeks before analysis (Fig. 4e, f). According to the regions of the hippocampus, adult neurogenesis may subserve different functions: new neurons born in the dorsal hippocampus influences cognitive information processing, whereas adult-born neurons of the ventral hippocampus regulate mood and stress response[29]. In the present study, the effects on UCMS microbiota on adult hippocampal neurogenesis were observed both in the dorsal and ventral regions of the hippocampus (Supplementary Fig. 5). Together, these data demonstrate that the decrease in hippocampal neurogenesis and depressive-like behaviors observed in recipients of UCMS microbiota can be rescued by selectively increasing the activity of the brain eCB system.

We next reasoned that, if UCMS microbiota induces paucity in serum levels of eCB precursors, the complementation of diet with

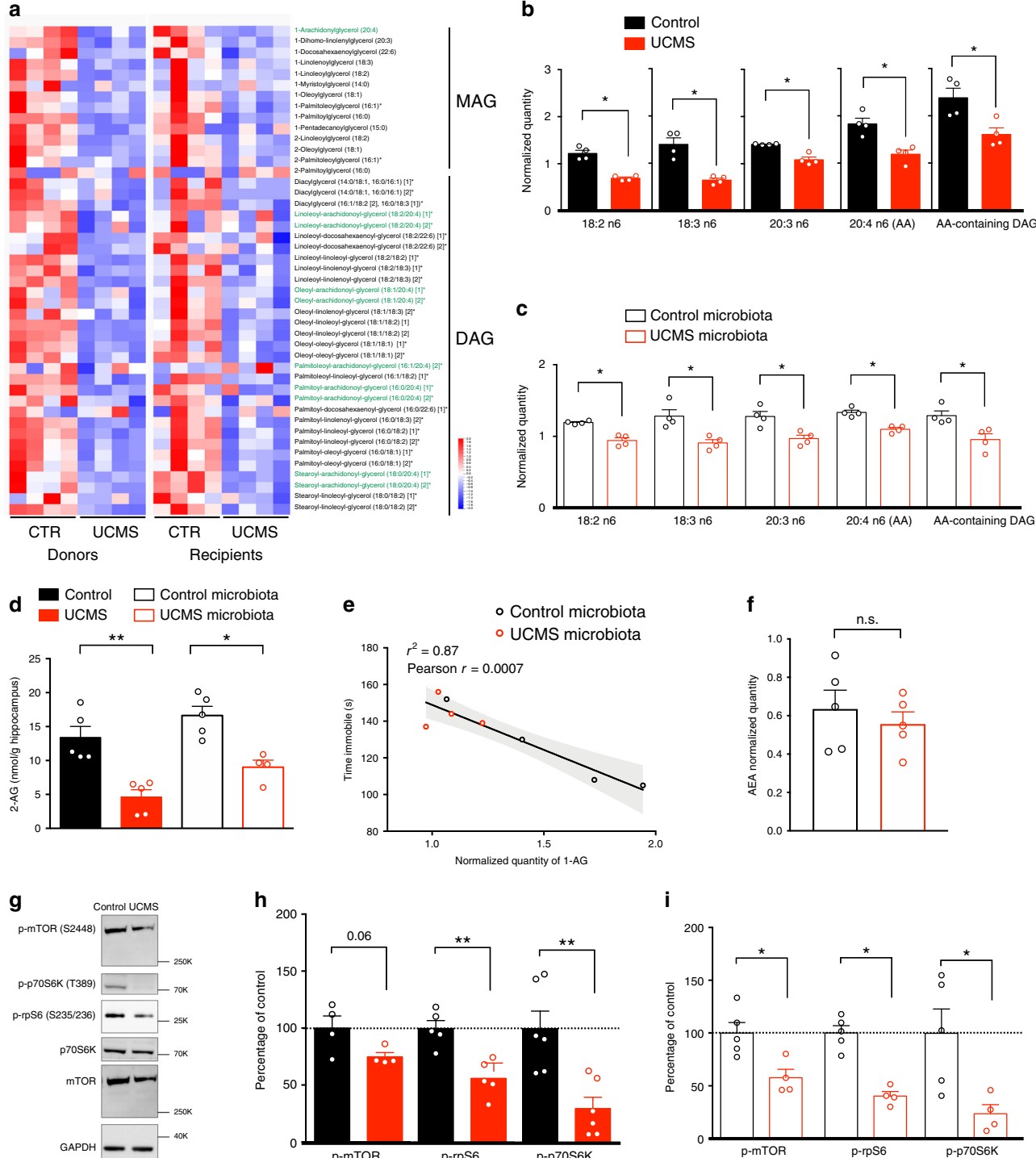

**Fig. 2 Microbiota from UCMS mice alters fatty acid metabolism and hippocampal eCB system. a** Heatmap of relative serum levels of monoacylglycerols (MAG) and diacylglycerols (DAG) in donor ($n = 4$/group) and recipient mice ($n = 4$/group) (z-scored). Arachidonic acid-containing MAG and DAG are highlighted in green. **b**, **c** Relative levels of fatty acid in the synthesis pathway of arachidonic acid (AA) and AA-containing DAG in donor mice ($n = 4$; **b**) and in recipient mice ($n = 4$, **c**). For **b**, **c**, data are represented as mean ± s.e.m. Statistical significance was calculated using Mann–Whitney test (*$P = 0.0286$). **d** Concentration of 2-AG in the hippocampus of donor ($n = 5$/group) and recipient mice ($n = 5$/group) was determined by targeted LC-MS (Control vs UCMS, $P = 0.0079$; Control microbiota- vs UCMS microbiota-recipient mice, $P = 0.0159$). **e** Correlation between serum quantity of 1-AG and time spent immobile in the tail suspension test (TST) in recipient mice ($n = 4$/group). Correlation was calculated using Pearson correlation factor $r$ ($r = 0.0007$). **f** Relative quantity of the endocannabinoid anandamide (AEA) in the hippocampus of recipient mice ($n = 5$/group, $P = 0.6905$). **g** Representative western blots for p-mTOR (S2448), p-rpS6 (S235/236), p-p70S6K (T389), mTOR, p70, and GAPDH in hippocampal protein extracts from donor mice. **h**, **i** Quantification of the phosphorylation of mTOR, rpS6, and p70S6K in protein extracts from the hippocampus of Control and UCMS donor mice (p-mTOR, $n = 4$, $P = 0.0571$; p-rpS6, $n = 5$, $P = 0.0079$; p-p70S6K, $n = 6$, $P = 0.0087$; **e**) and Control microbiota- ($n = 5$) and UCMS microbiota-recipient mice ($n = 4$) (p-mTOR, $P = 0.0317$; p-rpS6, $P = 0.0159$; p-p70S6K, $P = 0.0317$; **f**). For **b**–**d**, **f**, **h**, **i**, data are represented as mean ± s.e.m. Statistical significance was calculated using Mann–Whitney test (*$P < 0.05$, **$P < 0.01$, two tailed). Source data are provided as a Source data file.

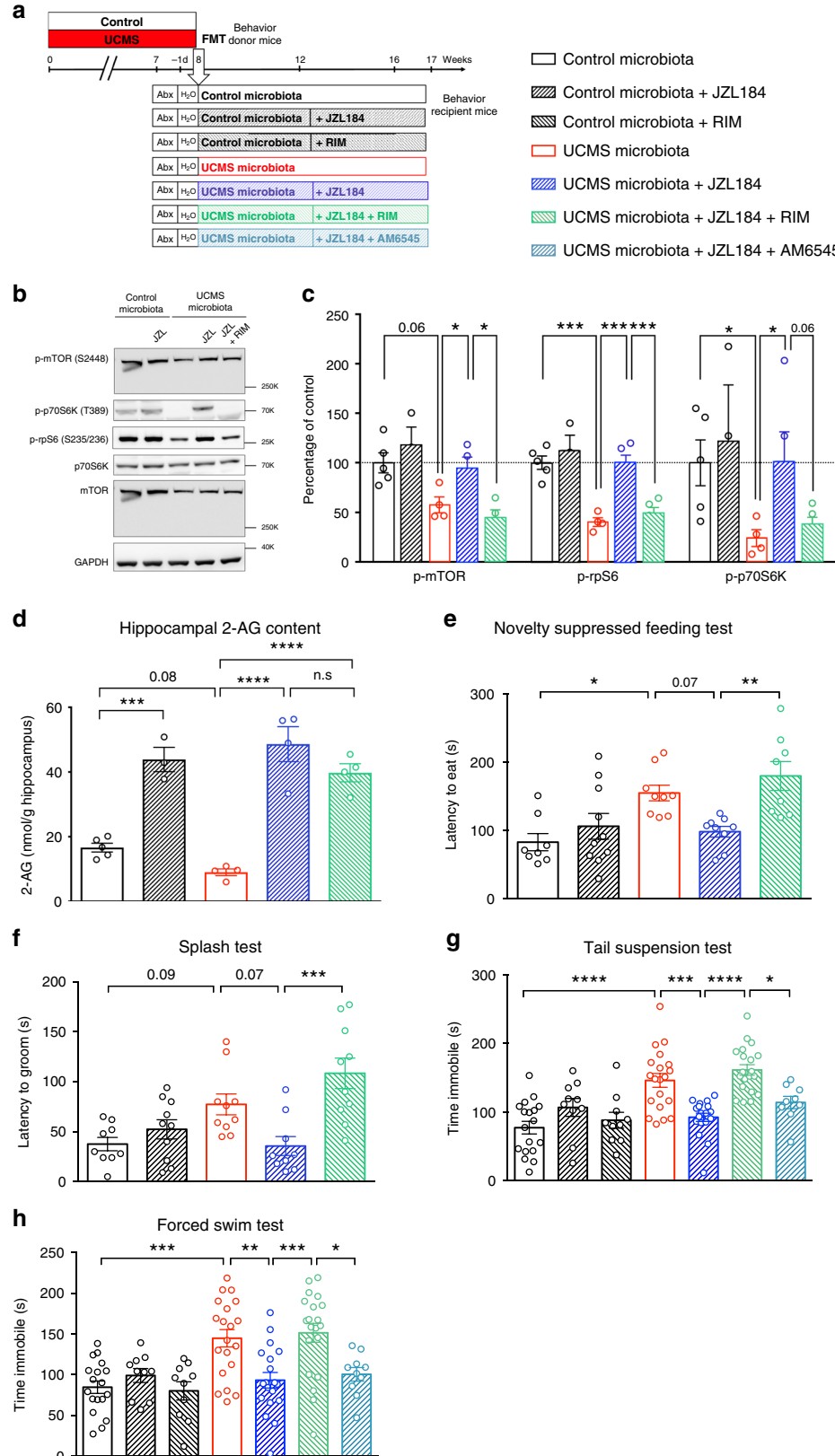

eCB precursors, such as AA, might restore the levels of 2-AG and restore normal behavior. Recipient mice of UCMS microbiota were given orally AA for 5 weeks starting 3 weeks after microbiota transfer (Fig. 5a). Remarkably, we observed that AA treatment restored normal levels of hippocampal 2-AG (Fig. 5b) and reversed the depressive-like behaviors induced by UCMS

microbiota (Fig. 5c, d). Furthermore, AA complementation also partially restored the production and the survival of hippocampal newborn neurons (Fig. 5e, f). In the hippocampus, we also observed a general tendency toward an increase of n-3 PUFA and n-6 PUFA (Supplementary Fig. 6A), as well as AEA, (Supplementary Fig. 6B) of UCMS microbiota-recipient mice

**Fig. 3 Restoration of the eCB pathway normalizes behavior in recipient mice. a** Experimental timeline of JZL184, rimonabant (RIM), and AM6545 treatment in recipient mice. Mice were injected intraperitoneally every 2 days, with vehicle alone, JZL184 (8 mg/kg), rimonabant (2 mg/kg), AM6545 (2 mg/kg), JZL184 + rimonabant, or JZL184 + AM6545. The treatment started 4 weeks after FMT and lasted for 5 weeks, until sacrifice. **b** Representative western blots for p-mTOR (S2448), p-rpS6 (S235/236), p-p70S6K (T389), mTOR, p70, and GAPDH in hippocampal protein extracts from recipient mice upon treatment with JZL184 or rimonabant. **c** Quantification of the phosphorylation of mTOR, rpS6, and p70S6K in hippocampal protein extracts from Control microbiota-recipient mice ($n = 5$), Control microbiota-recipient mice treated with JZL184 ($n = 3$), UCMS microbiota-recipient mice ($n = 4$), UCMS microbiota-recipient mice treated with JZL184 ($n = 5$, except for p-mTOR with $n = 4$), and UCMS microbiota-recipient mice treated with JZL184 and rimonabant ($n = 5$, except for p-mTOR with $n = 4$). Control microbiota and UCMS microbiota groups are the same as in Fig. 2i. (p-mTOR: Control vs UCMS-recipient mice, $P = 0.0317$; UCMS-recipient mice vs UCMS-recipient mice + JZL184, $P = 0.0571$; UCMS-recipient mice + JZL184 vs UCMS-recipient mice + JZL184 + RIM, $P = 0.0286$; p-rpS6: Control vs UCMS-recipient mice, $P = 0.0159$; UCMS-recipient mice vs UCMS-recipient mice + JZL184, $P = 0.0159$; UCMS-recipient mice + JZL184 vs UCMS-recipient mice + JZL184 + RIM, $P = 0.0079$; p-p70S6K: Control vs UCMS-recipient mice, $P = 0.0317$; UCMS-recipient mice vs UCMS-recipient mice + JZL184, $P = 0.0317$; UCMS-recipient mice + JZL184 vs UCMS-recipient mice + JZL184 + RIM, $P = 0.0556$). **d** Concentration of 2-AG in the hippocampus of Control microbiota-recipient mice ($n = 5$), Control microbiota-recipient mice treated with JZL184 ($n = 3$), UCMS microbiota-recipient mice ($n = 4$), UCMS microbiota-recipient mice treated with JZL184 ($n = 4$), and UCMS microbiota-recipient mice treated with JZL184 and rimonabant ($n = 4$), as determined by targeted LC-MS. Control microbiota and UCMS microbiota groups are the same as in Fig. 2d. (Control microbiota-recipient vs Control microbiota-recipient mice + JZL184, $P = 0.0002$; Control microbiota-recipient vs UCMS microbiota-recipient mice, $P = 0.0872$; UCMS microbiota-recipient vs UCMS microbiota-recipient mice + JZL184, $P < 0.0001$; UCMS microbiota-recipient vs UCMS microbiota-recipient mice + JZL184 + RIM, $P < 0.0001$; UCMS microbiota-recipient + JZL184 vs UCMS microbiota-recipient mice + JZL184 + RIM, $P = 0.3037$). **e** Latency to eat in a novel environment in the novelty suppressed feeding test for Control microbiota-recipient mice ($n = 8$), Control microbiota-recipient mice treated with JZL184 ($n = 10$), UCMS microbiota-recipient mice ($n = 9$), UCMS microbiota-recipient mice treated with JZL184 ($n = 9$), and UCMS microbiota-recipient mice treated with JZL184 and rimonabant ($n = 8$). (Control microbiota- vs UCMS microbiota-recipient mice, $P = 0.0179$; UCMS microbiota-recipient mice vs UCMS microbiota-recipient mice + JZL184, $P = 0.0796$; UCMS microbiota-recipient mice + JZL184 vs UCMS microbiota-recipient mice + JZL184 + RIM, $P = 0.0054$). **f** Latency to groom in the splash test for Control microbiota-recipient mice ($n = 9$), Control microbiota-recipient mice treated with JZL184 ($n = 10$), UCMS microbiota-recipient mice ($n = 10$), UCMS microbiota-recipient mice treated with JZL184 ($n = 9$), and UCMS microbiota-recipient mice treated with JZL184 and rimonabant ($n = 10$). (Control microbiota- vs UCMS microbiota-recipient mice, $P = 0.0946$; UCMS microbiota-recipient mice vs UCMS microbiota-recipient mice + JZL184, $P = 0.0721$; UCMS microbiota-recipient mice + JZL184 vs UCMS microbiota-recipient mice + JZL184 + RIM, $P = 0.0003$). **g** Time spent immobile in the tail suspension test for Control microbiota-recipient mice ($n = 18$), Control microbiota-recipient mice treated with JZL184 ($n = 10$), Control microbiota-recipient mice treated with rimonabant ($n = 10$), UCMS microbiota-recipient mice ($n = 20$), UCMS microbiota-recipient mice treated with JZL184 ($n = 19$), UCMS microbiota-recipient mice treated with JZL184 and rimonabant ($n = 20$), and UCMS microbiota-recipient mice treated with JZL184 and AM6545 ($n = 9$). (Control microbiota- vs UCMS microbiota-recipient mice, $P < 0.0001$; UCMS microbiota-recipient mice vs UCMS microbiota-recipient mice + JZL184, $P = 0.0002$; UCMS microbiota-recipient mice + JZL184 vs UCMS microbiota-recipient mice + JZL184 + RIM, $P < 0.0001$; UCMS microbiota-recipient mice + JZL184 + RIM vs UCMS microbiota-recipient mice + JZL184 + AM6545, $P = 0.0266$). **h** Time spent immobile in the forced swim test for Control microbiota-recipient mice ($n = 18$), Control microbiota-recipient mice treated with JZL184 ($n = 10$), Control microbiota-recipient mice treated with rimonabant ($n = 10$), UCMS microbiota-recipient mice ($n = 20$), UCMS microbiota-recipient mice treated with JZL184 ($n = 19$), UCMS microbiota-recipient mice treated with JZL184 and rimonabant ($n = 20$), and UCMS microbiota-recipient mice treated with JZL184 and AM6545 ($n = 10$). (Control microbiota- vs UCMS microbiota-recipient mice, $P = 0.0003$; UCMS microbiota-recipient mice vs UCMS microbiota-recipient mice + JZL184, $P = 0.0028$; UCMS microbiota-recipient mice + JZL184 vs UCMS microbiota-recipient mice + JZL184 + RIM, $P = 0.0004$; UCMS microbiota-recipient mice + JZL184 + RIM vs UCMS microbiota-recipient mice + JZL184 + AM6545, $P = 0.0276$). Data are represented as mean ± s.e.m. For **c–h**, statistical significance was calculated using one-way ANOVA with Tukey's multiple comparisons test (*$P < 0.05$, **$P < 0.01$, ***$P < 0.001$, ****$P < 0.0001$). Source data are provided as a Source data file.

supplemented with AA, compared to UCMS microbiota-recipient mice. We concluded that the recovery of adult neurogenesis and behaviors after AA complementation was associated with an increase in hippocampal eCBs—both 2-AG and AEA—and PUFAs.

**Complementation with *L. plantarum*$^{WJL}$ normalizes neurogenesis and behavior.** We next investigated how UCMS affected the composition of the microbiota that was responsible for the observed cellular and behavioral impairments in recipient mice. The composition of the fecal microbiota was determined by sequencing of 16S rDNA. Analysis of bacterial families revealed significant modifications in the microbiota of UCMS mice, as compared to the microbiota of control mice raised in separate cages (Fig. 5g), while the total number of species (alpha diversity) did not vary significantly (Fig. 5h). In-depth analysis of bacterial families showed an increase in *Ruminococcaceae*, and *Porphyromonodaceae*, as well as a decrease in *Lactobacillaceae* in UCMS mice (Fig. 5g and Supplementary Fig. 7). These results are in agreement with recent studies reporting an association between low frequencies of *Lactobacilli* and stress in mice[30–32] or depression in patients[33]. Importantly, the differences in microbiota composition between recipient mice of UCMS and control

microbiota were maintained 8 weeks after transfer (Fig. 5g), in particular the decrease in *Lactobacillaceae* (Fig. 5g and Supplementary Fig. 7), while the total number of species (alpha diversity) did not vary (Fig. 5h).

Since the frequencies of *Lactobacillaceae* were decreased in UCMS microbiota when compared to control microbiota (Fig. 5g), we tested whether complementation of UCMS microbiota with *Lactobacillaceae* restore behaviors and neurogenesis levels in recipients of UCMS microbiota. To this end, the microbiota of recipients was complemented with a strain of *Lactobacillus plantarum* ($Lp^{WJL}$) shown to modulate the host's lipid composition[34,35], to stimulate juvenile growth[36], and to influence affective behavior in mice[37]. Recipient mice of UCMS microbiota were given orally $Lp^{WJL}$ for 5 weeks starting 3 weeks after microbiota transfer (Fig. 5a). We observed that $Lp^{WJL}$ restored normal levels of hippocampal 2-AG (Fig. 5b), reversed the depressive-like behaviors induced by UCMS microbiota (Fig. 5c, d), and partially restored the production and the survival of hippocampal newborn neurons (Fig. 5e, f). In the hippocampus, we also observed an increase of n-3 and n-6 PUFA (Supplementary Fig. 6A) and AEA levels (Supplementary Fig. 6B) in UCMS microbiota-recipient mice supplemented with $Lp^{WJL}$. We concluded that the recovery of adult neurogenesis and behaviors after

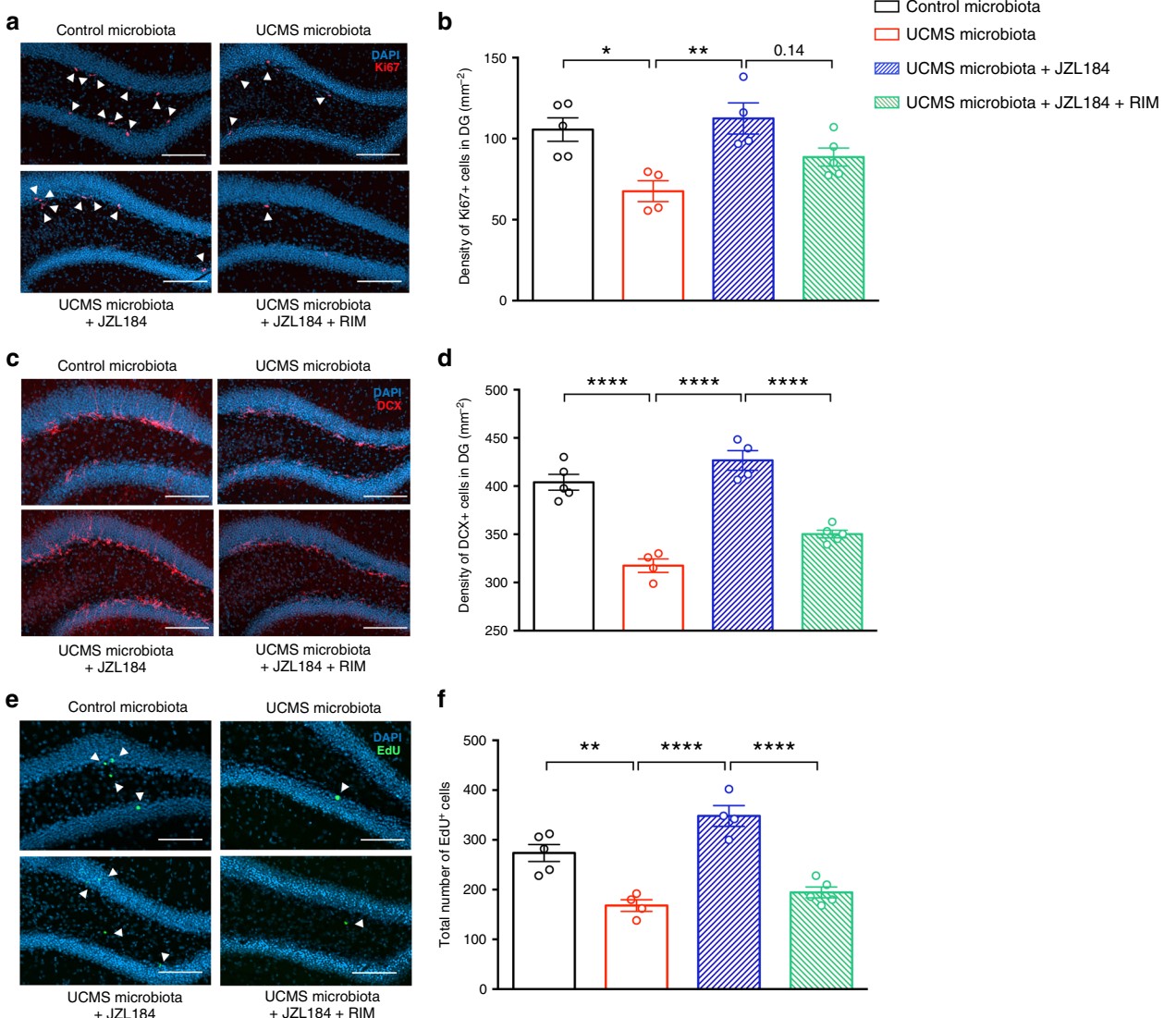

**Fig. 4 Restoration of the eCB pathway normalizes hippocampal neurogenesis. a** Representative images of Ki67 staining (red) in the DG of the hippocampus, counterstained with DAPI (blue). **b** Quantitative evaluation of the density of Ki67+ cells for Control microbiota-recipient mice ($n = 5$), UCMS microbiota-recipient mice ($n = 4$), UCMS microbiota-recipient mice treated with JZL184 ($n = 4$), and UCMS microbiota-recipient mice treated with JZL184 and rimonabant ($n = 5$). (Control microbiota- vs UCMS microbiota-recipient mice, $P = 0.0111$; UCMS microbiota-recipient mice vs UCMS microbiota-recipient mice + JZL184, $P = 0.0048$; UCMS microbiota-recipient mice + JZL184 vs UCMS microbiota-recipient mice + JZL184 + RIM, $P = 0.1402$). **c** Representative images of DCX staining (red) in the DG of the hippocampus, counterstained with DAPI (blue). **d** Quantitative evaluation of the density of DCX+ cells for Control microbiota-recipient mice ($n = 5$), UCMS microbiota-recipient mice ($n = 4$), UCMS microbiota-recipient mice treated with JZL184 ($n = 4$), and UCMS microbiota-recipient mice treated with JZL184 and rimonabant ($n = 5$). (Control microbiota- vs UCMS microbiota-recipient mice, $P < 0.0001$; UCMS microbiota-recipient mice vs UCMS microbiota-recipient mice + JZL184, $P < 0.0001$; UCMS microbiota-recipient mice + JZL184 vs UCMS microbiota-recipient mice + JZL184 + RIM, $P < 0.0001$). **e** Representative images of EdU staining (green) in the DG of the hippocampus, counterstained with DAPI (blue). **f** Quantitative evaluation of total number of EdU+ cells for Control microbiota-recipient mice ($n = 5$), UCMS microbiota-recipient mice ($n = 4$), UCMS microbiota-recipient mice treated with JZL184 ($n = 4$), and UCMS microbiota-recipient mice treated with JZL184 and rimonabant ($n = 5$). (Control microbiota- vs UCMS microbiota-recipient mice, $P = 0.0014$; UCMS microbiota-recipient mice vs UCMS microbiota-recipient mice + JZL184, $P < 0.0001$; UCMS microbiota-recipient mice + JZL184 vs UCMS microbiota-recipient mice + JZL184 + RIM, $P < 0.0001$). Scale bars: 100 μm. Data are represented as mean ± s.e.m. Statistical significance was calculated using one-way ANOVA with Tukey's multiple comparisons test (*$P < 0.05$, **$P < 0.01$, ****$P < 0.0001$). Source data are provided as a Source data file.

$Lp^{WJL}$ supplementation was associated with an increase in hippocampal eCBs—both 2-AG and AEA—and PUFAs.

## Discussion

In the present study, we have explored the mechanisms by which gut microbiota dysbiosis contributes to brain dysfunctions and behavioral abnormalities associated with depressive-like states. Chronic stress is recognized as a major risk factor for depression and most animal models of depressive-like behaviors rely on chronic stress or manipulation of the stress-sensitive brain circuits[38]. Using UCMS as a mouse model of depression, we showed that, upon transplantation to naive hosts, the microbiota from UCMS mice reduced adult hippocampal neurogenesis and induced depressive-like behaviors.

Searching for mechanistic explanations of these dysfunctions, we found that UCMS microbiota alters the fatty acid metabolism

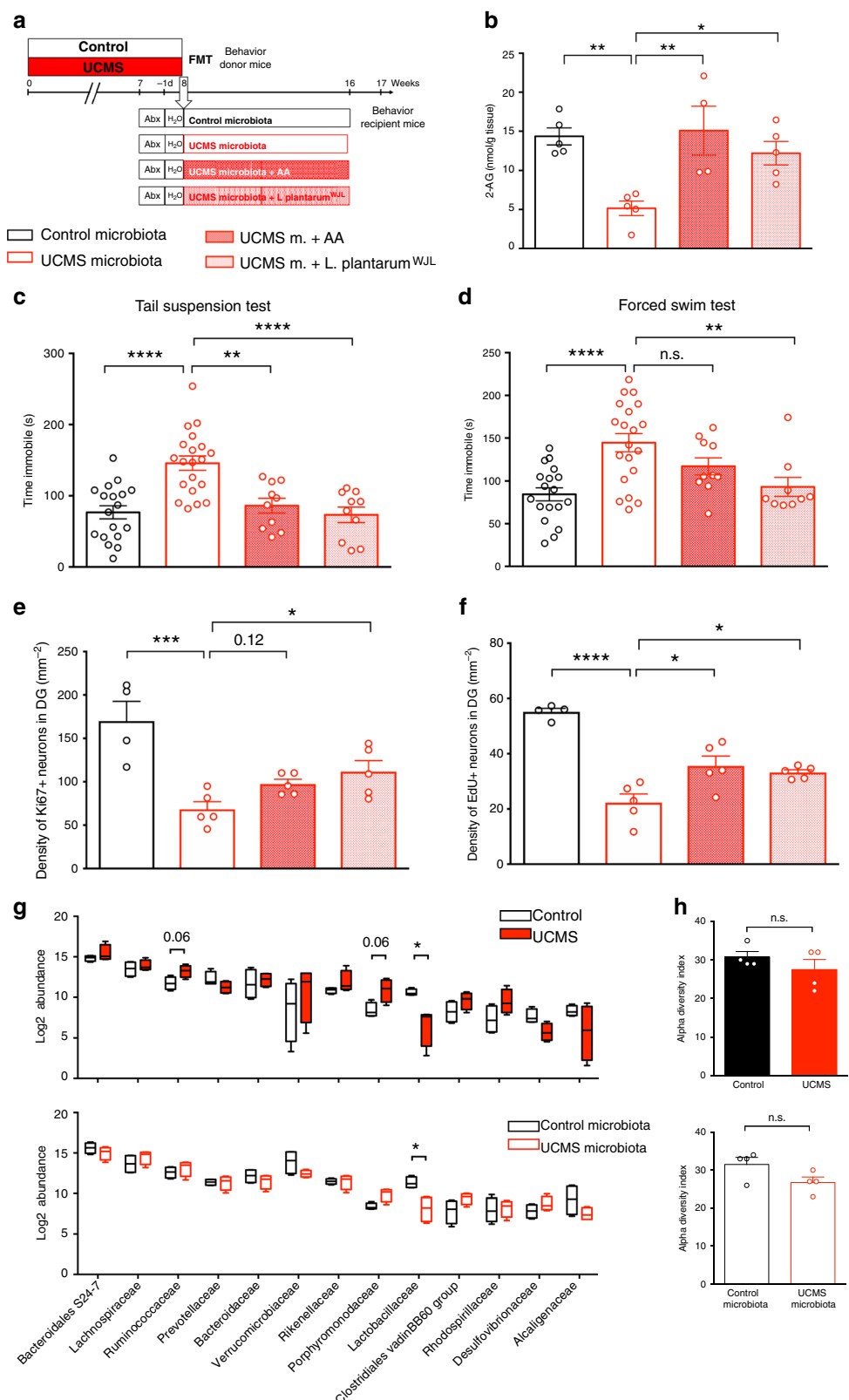

of the host, leading to paucity in precursors of the eCB system, such as AA, reduced production of the eCB 2-AG in the hippocampus, and diminished signaling in the hippocampal eCB system. Restoration of normal eCB signaling levels in mice recipient of UCMS microbiota after blocking the 2-AG-degrading enzyme or after complementation of the diet with the 2-AG precursor AA both restored adult neurogenesis and behaviors. Finally, UCMS-induced perturbations of the gut bacterial composition were characterized by loss of *Lactobacillaceae*, an alteration that was maintained after microbiota transplantation to naive hosts. The mere complementation of the UCMS recipients' microbiota with *L. plantarum Lp*[WJL] was sufficient to increase the levels of 2-AG in the hippocampus and restore affective behaviors and adult hippocampal neurogenesis. It is noteworthy that the positive

**Fig. 5 AA or $Lp^{WJL}$ complementation are sufficient to normalize adult neurogenesis and behavior. a** Experimental timeline of arachidonic acid (AA) and *L. plantarum* treatment in recipient mice. Mice were fed every 2 days through oral gavage with 8 mg of AA/mouse/day. Mice were supplemented by oral feeding 5 days a week with $2 \times 10^8$ CFU diluted in 200 µl of PBS. UCMS microbiota-recipient mice were oral fed with PBS as control. **b** Concentration of 2-AG in the hippocampus for Control microbiota ($n = 5$), UCMS microbiota ($n = 5$), UCMS microbiota complemented with AA ($n = 4$), and UCMS microbiota complemented with $Lp^{WJL}$ ($n = 5$), as determined by targeted LC-MS (Control microbiota- vs UCMS microbiota-recipient mice, $P = 0.0062$; UCMS microbiota-recipient mice vs UCMS microbiota-recipient mice + AA, $P = 0.0053$; UCMS microbiota-recipient mice vs UCMS microbiota-recipient mice + $Lp^{WJL}$, $P = 0.0371$). **c** Time spent immobile in the tail suspension test for Control microbiota-recipient mice ($n = 18$), UCMS microbiota-recipient mice ($n = 20$), UCMS microbiota-recipient mice complemented with AA ($n = 10$), and UCMS microbiota-recipient mice complemented with $Lp^{WJL}$ ($n = 10$) (Control microbiota- vs UCMS microbiota-recipient mice, $P < 0.0001$; UCMS microbiota-recipient mice vs UCMS microbiota-recipient mice + AA, $P = 0.0015$; UCMS microbiota-recipient mice vs UCMS microbiota-recipient mice + $Lp^{WJL}$, $P < 0.0001$). **d** Time spent immobile in the forced swim test for Control microbiota-recipient mice ($n = 18$), UCMS microbiota-recipient mice ($n = 20$), UCMS microbiota-recipient mice complemented with AA ($n = 10$), and UCMS microbiota-recipient mice complemented with $Lp^{WJL}$ ($n = 9$) (Control microbiota- vs UCMS microbiota-recipient mice, $P < 0.0001$; UCMS microbiota-recipient mice vs UCMS microbiota-recipient mice + AA, $P = 0.2690$; UCMS microbiota-recipient mice vs UCMS microbiota-recipient mice + $Lp^{WJL}$, $P = 0.0083$). Data are represented as mean ± s.e.m. Statistical significance was calculated using one-way ANOVA with Tukey's multiple comparisons test (**$P < 0.01$, ****$P < 0.0001$). **e** Quantitative evaluation of the density of Ki67+ cells for Control microbiota-recipient mice ($n = 4$), UCMS microbiota-recipient mice ($n = 5$), UCMS microbiota-recipient mice complemented with AA ($n = 5$), and UCMS microbiota-recipient mice complemented with $Lp^{WJL}$ ($n = 5$) (Control microbiota- vs UCMS microbiota-recipient mice, $P = 0.0003$; UCMS microbiota-recipient mice vs UCMS microbiota-recipient mice + AA, $P = 0.1175$; UCMS microbiota-recipient mice vs UCMS microbiota-recipient mice + $Lp^{WJL}$, $P = 0.0258$). **f** Quantitative evaluation of the density of EdU+ cells for Control microbiota-recipient mice ($n = 4$), UCMS microbiota-recipient mice ($n = 5$), UCMS microbiota-recipient mice complemented with AA ($n = 5$), UCMS microbiota-recipient mice complemented with $Lp^{WJL}$ ($n = 5$) (Control microbiota- vs UCMS microbiota-recipient mice, $P < 0.0001$; UCMS microbiota-recipient mice vs UCMS microbiota-recipient mice + AA, $P = 0.0113$; UCMS microbiota-recipient mice vs UCMS microbiota-recipient mice + $Lp^{WJL}$, $P = 0.0394$). **g** 16S rDNA of the fecal microbiota of donor mice at the end of the 8 weeks UCMS protocol ($n = 4$/group, top) or recipients mice after 8 weeks in isolators ($n = 4$/group, bottom) was sequenced and analyzed by principal component analysis (PCA) at the level of bacterial families for the relative abundance of bacterial families. Data are represented as boxplots, with median, minima, and maxima. Statistical significance was calculated using Mann–Whitney test (top, *Ruminococcaceae*, $P = 0.0571$; *Porphyromonadaceae*, $P = 0.0571$; *Lactobacillaceae*, $P = 0.0286$; bottom, *Lactobacillaceae*, $P = 0.0286$, two tailed). **h** Alpha diversity for donors ($P = 0.6857$, top, two tailed) and recipients ($P = 0.2286$, two tailed). Data are represented as mean ± s.e.m. Statistical significance was calculated using Mann–Whitney test. For **b–f**, data are represented as mean ± s.e.m. Statistical significance was calculated using one-way ANOVA with Tukey's multiple comparisons test (*$P < 0.05$, ***$P < 0.0005$, ****$P < 0.0001$). Source data are provided as a Source data file.

effects of AA and $Lp^{WJL}$ may have broad effect on the CB1 signaling pathway since both AA and $Lp^{WJL}$ complementation was able to increase both hippocampal 2-AG and AEA levels.

The eCB system has been reported to regulate mood, emotions, and responses to stress through activation of the cannabinoid receptor CB1. For instance, the CB1 receptor antagonist rimonabant, initially prescribed for the treatment of obesity and associated metabolic disorders, increases the incidence of depressive symptoms[39]. Furthermore, a higher frequency in a mutant allele for the CB1 receptor gene *CNR1* is observed in depressed patients[40]. In contrast, cannabis (that includes the eCB ligand delta-9 tetrahydrocannabinol) improves mood in humans[41], and low doses of synthetic CB1 agonists—but not high doses—produce anxiolytic- and antidepressant-like effects in animal models[42]. In particular, chronic stress has been showed to decrease eCB signaling in the brain[43]. Here we show that the intestinal microbiota is sufficient to initiate a pathological feed-forward loop for depressive disorders by impairing the eCB system in the hippocampus, a brain region strongly involved in the development of depressive symptoms. In the hippocampus, we observed a specific decrease in 2-AG—but not in AEA—in mice receiving UCMS-derived microbiota. This result is consistent with clinical observations reporting low serum levels of 2-AG in patients suffering from depression, posttraumatic stress disorder, or chronic stress but not of the other main eCB ligand AEA[44–46]. Previous studies have shown that the reduction in hippocampal CB1 signaling—but not a complete blockade—induces depressive-like behaviors[47]. In the hippocampus, CB1 receptors, together with other external signals, constitute a potent activator of mTOR signaling. Interestingly, deficits in mTOR signaling have been observed in different studies on postmortem brains of depressed patients and is considered as an emerging pathway for antidepressant treatments[48].

The eCB system exerts its pleiotropic effects through multiple neuronal processes, including, but not limited to, adult hippocampal neurogenesis. The eCB system is known to regulate adult neurogenesis via the CB1 receptor[49] expressed by neural progenitor cells[26,28]. CB1-deficient mice show impaired neural progenitor proliferation, self-renewal, and neurosphere generation[26], whereas CB1 receptor agonists increase neurogenesis[42,50]. In addition to this neurogenic effect occurring in the hippocampus, other CB1 receptor-dependent processes might contribute to the pathophysiology of our microbiota-induced depression. Further studies should be conducted to test whether other brain targets of eCB signaling are equally affected by microbiota dysbiosis.

It has been reported previously that the microbiota modulates the activity of the eCB system in the gut[51]. In the present study, we further demonstrate that the dysbiotic gut microbiota from UCMS mice is sufficient to induce dysregulation of the eCB system in the brain. We report that this dysregulation originates from a systemic decrease in eCB precursors. Modifications in gut microbiota composition following chronic stress has been extensively reported[30,32]. In particular, low frequencies of *Lactobacillaceae* are correlated with stress levels in mouse models[30–32]. Dysbiosis of the gut microbiota and low *Lactobacilli* frequency have also been detected in depressed patients[12,13,16]. The mechanisms by which chronic stress induces intestine dysbiosis has not yet been figured out and may involve subtle changes of the gut homeostasis, alterations of the enteric nervous and immune systems, or changes in some metabolic pathways. Importantly, transplantation of the dysbiotic microbiota from these patients into germ-free rats induces depressive- or anxiety-like behaviors in the recipients[16]. In line with these results, a probiotic treatment with *Lactobacilli* ameliorates depressive- and anxiety-like behaviors in mice[32]. Gut microbiota also modulates adult neurogenesis[52,53] and a *Lactobacillus* strain has been shown to promote the survival of hippocampal neuronal progenitor[52]. Numerous studies have shown that *Lactobacilli* treatment, as well as the administration of other probiotics, are beneficial in significantly lowering depression and anxiety scores in patients[54,55],

although the relative efficacy of probiotics compared to anti-depressants is still a matter of debate. *L. plantarum* in particular was recently shown to alleviate stress and anxiety[56]. The positive influence of *Lactobacillus* strain administration on mood may rely on multiple mechanisms, including the regulation of kynurenine production, HPA axis, and immunomodulation. Our study demonstrating the beneficial effects of *L. plantarum*[WJL] to complement a maladaptive microbiota adds to several emerging evidence showing an antidepressant effects of probiotics in major depression[57]. We have found that one of the mechanisms by which *Lactobacilli* promotes these effects is through regulation of the bioavailability of eCB precursors.

A major finding of our study is that recipients of UCMS microbiota developed an altered fatty acid metabolism characterized by deficiency in MAG, DAG, and fatty acids. Serum levels of MAG, DAG, and PUFAs were inversely correlated with the severity of depressive-like behaviors. Further studies should clarify whether serum levels of fatty acid could be considered as earlier biomarker for mood disorders. It has been reported that nutritional n-3 PUFA deficiency abolishes eCB-mediated neuronal functions[58] and, conversely, that n-3 PUFA dietary supplementation reverses some aspects of UCMS-induced depressive-like behaviors in mice[58,59]. We speculate that UCMS microbiota (1) promotes the degradation of PUFA, (2) alters the absorption of these fatty acids (for instance, by modifying the excretion of bile acids, as observed upon chronic stress[60]), or (3) disrupt the regulation of lipid synthesis (as observed in rodents upon chronic stress[61]). The mechanisms by which gut microbiota modulates the host's fatty acid metabolism has been partially investigated in several animal models. *Lactobacilli* species can regulate intestinal absorption and metabolism of fatty acids in the zebrafish[62,63]. In mammals, *Lactobacilli* are more prominent in the small intestine, the primary site of lipid absorption[64]. Studies in rodents have shown that *Lactobacilli* species modulate lipid metabolism[65,66]. In addition, *Lactobacilli* may indirectly influence lipid absorption by modulating intestinal transit. Specifically, *L. plantarum* modulates the host's lipid composition by reducing the level of serum triglycerides in the context of high-fat diet[34,67]. Furthermore, in humans, *L. plantarum* is associated with lower levels of cholesterol[35]. Altogether, *L. plantarum* is thought to regulate fatty acid metabolism and modifies fatty acid composition of the host[66], even though we cannot rule out that other *Lactobacilli* species may be effective at normalizing the neurobehavioral phenotype observed in UCMS microbiota-recipient mice by regulating host's lipid composition.

In sum, our data show that microbiota dysbiosis induced by chronic stress affects lipid metabolism and the generation of eCBs, leading to decreased signaling in the eCB system and reduced adult neurogenesis in the hippocampus. This might be the pathway, at least in part, that links microbiota dysbiosis to mood disorders, which in turn, may affect the composition of the gut microbiota through physiological adjustments and modulation of the immune system. Because we were able to interrupt this pathological feed-forward loop by administering AA or a *Lactobacillus* probiotic strain, our study supports the concept that dietary or probiotic interventions might be effective levers in the therapeutic arsenal to fight stress-associated depressive syndromes.

## Methods
**Mice**. Adult male C57BL/6J mice (8–10 weeks old) were purchased from Janvier laboratories (St Berthevin, France) and maintained under SPF conditions at the Institut Pasteur animal care facility. Germ-free C57BL/6J mice were generated at the Gnotobiology Platform of the Institut Pasteur and routinely monitored for sterility. Mice were provided with food and water ad libitum and housed under a strict 12-h light–dark cycle. All animal experiments were approved by the

committee on animal experimentation of the Institut Pasteur (project CETEA #2013-0062 and #2016-0023) and by the French Ministry of Research.

**FMT protocol**. Recipient mice were given a combination of antibiotics derived from Le Roy et al.[68], namely, vancomycin (0.5 g/l), ampicillin (1 g/l), streptomycin (5 g/l), colistin (1 g/l), and metronidazole (0.5 g/l), in their drinking water for 6 consecutive days. All antibiotics were obtained from Sigma-Aldrich (St Quentin Fallavier, France). Twenty-four hours later, animals were colonized via two rounds of oral gavage with microbiota, separated 3 days apart, and kept in separate sterile isolators. Fresh fecal pellets were collected directly from the rectum of donor mice and 100 mg (about 5–6 fecal pellets) were homogenized in 1 ml sterile phosphate-buffered saline (PBS). Homogenates were then passed through a 20-μm pore nylon filter to remove large particulate and fibrous matter. The solution was collected and 200 μl was administered by oral gavage to recipient mice, within 15 min to minimize changes in microbial contents. Recipient animals were left in the isolators for 8 weeks before any analysis. Such 8-week delay between FMT and analysis was motivated by the fact that the C57BL/6J mouse line is considered relatively stress resistant and requires at least 8 weeks to display depression-like symptoms[23].

**UCMS protocol**. After 1 week of habituation to the Institut Pasteur facility upon arrival, mice were subjected to various and repeated unpredictable stressors several times a day for 8 weeks. During exposure to stressors, mice of the UCMS group were housed in a separate room. The stressors included altered cage bedding (recurrent change of bedding, wet bedding, no bedding), cage tilting (45°), foreign odor (new cage impregnated with foreign mouse urine), restraint (30 min in a clean 50 ml conical tube with pierced holes for ventilation), and altered light/dark cycle. On average, two stressors were administered per day. The timeline of the stressor exposure is described in Supplementary Table S1. For stressed animals, cages were changed after "wet bedding' and "no bedding' stressors. Unstressed controls were handled only for injections, cage changes, and behavioral tests.

**CB1 antagonists and JZL184 treatment**. JZL184, rimonabant, and AM6545 were purchased from Cayman Chemicals (Bertin Technologies, Montigny-le-Bretonneux, France). The drugs were dissolved in a vehicle containing a 1:1:18 mixture of ethanol, kolliphor, and saline and injected intraperitoneally (i.p.) at a volume of 10 μl/g bodyweight every 2 days. Mice were injected with vehicle alone, JZL184 (8 mg/kg), rimonabant (2 mg/kg), AM6545 (2 mg/kg), JZL184 + rimonabant, or JZL184 + AM6545. The dose and treatment time of drug administration, alone or in combination, were chosen based on previous studies showing that JZL184 irreversibly inhibits the MAGL and produces at least twofold increase in 2-AG levels in the brain at a dose of 8 mg/kg when dissolved in the vehicle used in this study[69,70]. Repeated administration of JZL184 at this low dose does not induce observable CB1 receptor desensitization or functional tolerance[71].

**AA and *Lactobacilli* complementation**. AA was purchased from Cayman Chemicals (Bertin Technologies). Mice were fed every 2 days through oral feeding gavage with 8 mg of AA/mouse/day. *L. plantarum Lp*[WJL] was kindly provided by Pr. François Leulier (ENS, Lyon, France), and mice were supplemented by oral feeding 5 days a week with $2 \times 10^8$ colony-forming units diluted in 200 μl of PBS. UCMS microbiota-recipient mice were free fed with only PBS as control.

**Microbial DNA extraction and 16S sequencing**. Total DNA was extracted from feces using the FastDNA Spin Kit, following the instructions of the manufacturer (MP Biomedicals). DNA concentrations were determined by spectrophotometry using a Nanodrop (Thermo Scientific). Microbial composition was assessed by 16S metagenomic analysis, performed on an Illumina MiSeq instrument using a v3 Reagent Kit. Libraries were prepared by following the Illumina "16S Metagenomic Sequencing Library Preparation" protocol (Part # 15044223 Rev. B) with the following primers: Forward—5'-TCGTCGGCAGCGTCAGATGTGTATAAGAGAC AGCCTACGGGNGG-CWGCAG-3'; Reverse—5'-GTCTCGTGGGCTCGGAG ATGTGTATAAGAGACAGGA-CTACHVGGGTATCTAATCC-3'. PCR amplification targeted the V3–V4 region of the 16s rDNA. Following purification, a second PCR amplification was performed to barcode samples with the Nextera XT Index Primers. Libraries were loaded onto a MiSeq instrument and sequencing was performed to generate $2 \times 300$ bp paired end reads. De-multiplexing of the sequencing samples was performed on the MiSeq and individual FASTQ files recovered for analysis.

**16S data analysis**. Sequences were clustered into OTUs (Operational Taxonomic Units) and annotated with the MASQUE pipeline (https://github.com/aghozlane/masque) as described[72]. OTU representative sequences assigned to the different taxonomic levels using RDP Seqmatch (RDP database, release 11, update 1)[73]. Relative abundance of each OTU and other taxonomic levels was calculated for each sample in order to consider different sampling levels across multiple individuals. After trimming, numbers of sequences clustered within each OTU (or other taxonomic levels) were converted to relative abundances. All the analysis were performed blinded to experimental conditions. Statistical analyses were performed with SHAMAN (shaman.c3bi.pasteur.fr) as described[74]. Briefly, the normalization of

OTU counts was performed at the OTU level using the DESeq2 normalization method. In SHAMAN, a generalized linear model was fitted and vectors of contrasts were defined to determine the significance in abundance variation between sample types. The resulting *P* values were adjusted for multiple testing according to the Benjamini and Hochberg procedure[75]. Principal coordinates analysis was performed with the *ade4* R package (v.1.7.6) using a Bray–Curtis dissimilarity matrix. Further statistical analysis was conducted using the Prism software (GraphPad, v6, San Diego, USA). Differences between two groups were assessed using Mann–Whitney test for family abundance.

**Gut permeability test**. This examination is based on the intestinal permeability to 4 kD fluorescent–dextran (Sigma-Aldrich). After 4 h of food withdrawal, mice were orally administered with FITC–dextran (0.6 g/kg body weight). After 1 h, 200 µl of blood was collected in Microvette® tube (Sarstedt, Marnay, France). The tubes were then centrifuged at $10,000 \times g$ for 5 min, at room temperature, to extract the serum. Collected sera were diluted with same volume of PBS and analyzed for FITC concentration at excitation wavelength of 485 nm and the emission wavelength of 535 nm. All the analysis was performed blinded to experimental conditions.

**Behavioral assays**. Anxiety and depressive-like behaviors were assessed at time points of interest. Mice were tested for LDB, splash test, novelty suppressed feeding, tail suspension test, and forced swim test, in that order. In order to limit the eventual microbiota divergence once the recipient mice were removed from the isolators, behavioral tests were performed within a week, with at least 24 h between each behavioral test. Order of passage between groups was randomized, and all the analysis was performed blinded to experimental conditions. Anxiety-like behaviors were evaluated in the LDB tests. Depressive-like behaviors were evaluated in the splash test, the novelty suppressed feeding test, the tail suspension test, and the forced swim test.

- *Light/dark box*. The test was conducted in a 44 cm × 21 cm × 21 cm Plexiglas box divided into dark and light compartments separated by an open door. The light in the light compartment was set up at 300 lux. Time spent in the light compartment and transitions between compartments during 10 min were video-tracked using the EthoVision XT 5.1 software (Noldus Information Technology).
- *Splash test*. The splash test consists of squirting a 10% sucrose solution on the dorsal coat of a mouse in its home cage. Because of its viscosity, the sucrose solution dirties the mouse fur and animals initiate grooming behavior. After applying sucrose solution, latency to grooming, frequency, and time spent grooming was recorded for a period of 6 min as an index of self-care and motivational behavior. The splash test, pharmacologically validated, demonstrates that UCMS decreases grooming behavior, a form of motivational behavior considered to parallel with some symptoms of depression, such as apathetic behavior[6,22,76].
- *Novelty suppressed feeding (NSF)*. The NSF was carried out similar to a published protocol[22]. Mice were deprived of food for 24 h before being placed in a novel environment, a white plastic box (50 cm × 50 cm × 20 cm) whose floor was covered with wooden bedding. A single food pellet (regular chow) was placed on a piece of filter paper (10 cm in diameter), positioned in the center of the container that was brightly illuminated (~500 lux). The mouse was placed in one corner of the box and the latency to feed was measured during 10 min. Feeding was defined as biting, not simply sniffing or touching the food. Immediately after the test, the animals were transferred into their home cage and the amount of food consumed over the subsequent 5-min period were measured as a control of feeding behavior.
- *Tail suspension test*. Mice were suspended by the tail using adhesive tape affixed 1 cm from the origin of the tail, on a metal rod under dim light conditions (~40 lux). The behavior of the animals was recorded by a video camera during a 5-min period and total immobility time was evaluated in a blind manner.
- *Forced swim test*. Mice were placed individually into plastic cylinders (19 cm diameter, 25 cm deep) filled to a depth of 18 cm with water (23–25 °C) under dim light conditions (~40 lux) for 5 min. The behavior of the animals was recorded by a video camera and immobility time was automatically evaluated using the EthoVision XT 5.1 software (Noldus Information Technology).

In both TST and FST, mice face an uncomfortable situation that they confront by attempting to move out of it and eventually surrender to.

**EdU labeling**. The study of proliferation and differentiation of neural stem cells in the DG was performed by incorporation of EdU (Click-iT EdU Imaging Kit; Molecular Probes) to allow the analysis of proliferation and differentiation. Mice received four i.p. injections (100 mg/kg), at 2 h intervals, on a single day, 4 weeks before perfusion, for the analysis of cell survival. EdU incorporation was visualized as described in the "Immunohistochemical analysis" section.

**Immunohistochemical analysis**. Mice were deeply anesthetized with sodium pentobarbital (i.p., 100 mg/kg, Sanofi) and perfused transcardially with a solution containing 0.9% NaCl and heparin (Sanofi-Synthelabo), followed by 4%

paraformaldehyde in phosphate buffer, pH 7.3. Brains were removed and postfixed by incubation in the same fixative at 4 °C overnight. Tissues were cryoprotected by incubation in 30% sucrose in PBS for 24 h. Immunostaining was performed on 40- or 60-µm-thick coronal brain sections obtained with a vibrating microtome (VT1000S, Leica). Nonspecific staining was blocked by 0.2% Triton, 4% bovine serum albumin (Sigma-Aldrich), and 2% goat serum and free-floating slices were then incubated with the following primary antibodies at 4 °C overnight: rabbit anti-DCX (Abcam, ab 18723), rabbit anti-Ki67 (Abcam, ab16667), and mouse anti-NeuN (Millipore, MAB377). Secondary antibodies (Alexa Fluor-conjugated secondary antibodies, Molecular Probes) were then incubated at room temperature. 4,6-Diamidino-2-phenylindole (1 µg/ml) was used as a nuclear stain. EdU was visualized using the Click-iT reaction coupled to an Alexa Fluor® azide following the instructions of the manufacturer (Molecular Probes).

**Image acquisition and quantification analysis**. Immunofluorescence was analyzed using an Apotome microscope (Apotome.2; Zeiss) with the Zen Imaging software (Zeiss), courtesy of Pr. Peduto. Quantification was performed using the Icy open source platform (http://www.icy.bioimageanalysis.org)[77]. The region of interest was defined as the granule cell layer (GCL) of the DG, and automatic detection of Ki67$^+$ and DCX$^+$ cells was performed using the spot detector tool. Values are expressed as the mean of total Ki67$^+$ or DCX$^+$ cell count per mm$^2$ in six slices per animal. All imaging and quantification were performed blinded to experimental conditions. For EdU analysis, positive cells were manually counted in the GCL of the DG. Total number was estimated by multiplying the total number of cells every sixth section by six.

**Western blotting**. Mice were deeply anesthetized with sodium pentobarbital (i.p. 100 mg/kg, Sanofi) and rapidly decapitated. The hippocampi were bilaterally dissected out and then homogenized in 0.2 ml lysis buffer (pH 7.5) containing 20 mM Tris-acetate, 150 mM NaCl, 50 mM NaF, 1 mM EDTA, 1% Triton-X100, 0.1% benzonase, protease inhibitors, and protein phosphatase inhibitors I and II (Sigma-Aldrich). After an incubation of 30 min on ice and centrifugation at $10,000 \times g$ for 10 min, total protein concentration of the supernatant was assayed by using the Bio-Rad Protein Assay Kit (Bio-Rad, Marnes-la-Coquette, France). Equal amounts of each protein sample were separated on NuPAGE Bis-Tris or Tris-Acetate gels and transferred to nitrocellulose or polyvinylidene difluoride membranes, respectively. Blots were blocked in blocking buffer containing 5% (w/v) milk and 0.1% (v/v) Tween-20 in Tris-buffered saline (TBS-T) for 1–2 h at room temperature and incubated overnight at 4 °C with antibodies against p-mTOR (S2448) (1:1000, Cell Signaling), mTOR (1:1000, Cell Signaling), p-p70S6K (T389) (1:500, R&D Systems), p-rpS6 (S235/236) (1:500, R&D Systems), or glyceraldehyde 3-phosphate dehydrogenase (1:1000, Cell Signaling) antibodies. Blots were washed 3 times with TBS-T and then probed with anti-rabbit IgG, horseradish peroxidase-linked antibody (1:3000, Cell Signaling) for 1 h at room temperature before being revealed using ECL Prime detection reagent (GE Healthcare) and chemiluminescence reading on a luminescent image analyzer (LAS-4000; Fujifilm). Immunoreactivity of western blots was quantified by densitometry using the ImageJ software (NIH, Bethesda) and the analysis was performed blinded to experimental conditions.

**Biochemical detection of hippocampal 2-AG**. Mice were deeply anesthetized with sodium pentobarbital (i.p. 100 mg/kg, Sanofi) and decapitated. The brain was immediately removed, and the hippocampi were dissected out and rapidly frozen on dry ice. 2-AG was extracted from the hippocampus as previously described[78]. Samples were weighed and placed into borosilicate glass culture tubes containing 2 ml of acetonitrile (ACN) with 186 pmol [$^2$H$_8$] 2-AG. They were homogenized using IKA homogenizer and kept overnight at −20 °C to precipitate proteins and subsequently centrifuged at $1500 \times g$ for 3 min. The supernatants were transferred to a new glass tube and evaporated to dryness under N$_2$ gas. The samples were resuspended in 500 µl of methanol to recapture any lipids adhering to the glass tube and dried again under N$_2$ gas. Dried lipid extracts were suspended in 50 µl of methanol and stored at −80 °C until analysis. The content of 2-AG was determined using isotope-dilution liquid chromatography–electrospray ionization tandem mass spectrometry (LC-MS/MS)[79] and the content of both 2-AG and 1(3)-AG isomers were pooled for quantification, given the potential isomerization between 1 (3)-AG and 2-AG following ACN precipitation. All the analysis was performed blinded to experimental conditions.

**Biochemical detection of total hippocampal PUFA and AEA**. Mice were deeply anesthetized with sodium pentobarbital (i.p. 100 mg/kg, Sanofi) and decapitated. The brain was immediately removed, and the hippocampi were dissected out and rapidly frozen on dry ice. PUFAs were obtained after hydrolysis of all lipid molecular species from mice brain hippocampus (total PUFA pool). Briefly, hippocampus samples were added to ACN according to each fresh weight, vortexed for 60 s, and then sonicated for 30 s using a sonication probe. A solution of ACN/HCl (4:1, v/v) was added to samples and placed in water bath at 90 °C during 2 h for hydrolysis before centrifugation at $20,000 \times g$ during 10 min. Supernatants were dried under a stream of nitrogen and resuspended in ACN solution according to their fresh weight before analysis by supercritical fluid chromatography coupled to mass spectrometry (SFC-MS). For AEA, a part of hippocampus samples in ACN

were vortexed for 60 s and then sonicated for 30 s using a sonication probe before centrifugation at $20,000 \times g$ during 10 min. Supernatants were dried under a stream of nitrogen and resuspended in ACN solution according their fresh weight before analysis by SFC-MS.

PUFAs and AEA were separated on a SFC (Acquity UPC2, Waters, France) using a Torus 1-Aminoanthracene $100 \times 3.0$ mm, 1.7 μm column (Waters, France) at 60 °C. Mobile phase A containing $CO_2$ and B containing MeOH and 0.2% formic acid were used for separation during a running time of 6.5 min with a flow rate at 1.8 ml/min. Wash weak and strong weak solutions are composed of Heptane/Isopropanol (90:10, v/v) and Isopropanol/Methanol/Chloroform (45:45:10, v/v/v) respectively. Finally, the make-up solution is composed of ACN. MS data were acquired in negative ionization mode on a Xevo TQ-MS triple quadrupole mass spectrometer (Waters, USA) controlled by the Masslynx software (version 4.1) with an ion spray voltage of 2500 V, a source, and a desolvation temperature set to 150 and 450 °C. Cone voltage was set to 30 V for PUFAs C18:2 and C18:3 and 25 V for PUFAs C20:5, C20:4, C22:6, deuterated C18:0 d4 as internal standard, and AEA. Multiple reaction monitoring (MRM) experiments were performed using parent–parent ion transitions for PUFAs with a collision energy (CE) at 5 eV while AEA analysis was performed by MRM experiment using transition 346 to 259 with a CE set to 15 eV. Following mean normalization, data were analyzed using one-way analysis of variance (ANOVA). All the quantification was performed blinded to experimental conditions.

**Metabolomics**. Blood were collected by cardiac puncture in Microvette® tubes (Sarstedt, Marnay, France) from behaviorally validated adult mice. The tubes were centrifuged at $10,000 \times g$ for 5 min, at room temperature, to extract the serum. Serum samples were then extracted and analyzed on gas chromatography tandem mass spectrometry (GC/MS), LC/MS, and LC/MS/MS platforms by Metabolon, Inc. (CA, USA), blinded to the experimental conditions. Protein fractions were removed by serial extractions with organic aqueous solvents, concentrated using a TurboVap system (Zymark) and vacuum dried. For LC/MS and LC/MS/MS, samples were reconstituted in acidic or basic LC-compatible solvents containing >11 injection standards and run on a Waters ACQUITY UPLC and Thermo-Finnigan LTQ mass spectrometer, with a linear ion-trap front-end and a Fourier transform ion cyclotron resonance mass spectrometer back-end. For GC/MS, samples were derivatized under dried nitrogen using bistrimethyl-silyl-trifluoroacetamide and analyzed on a Thermo-Finnigan Trace DSQ fast-scanning single-quadrupole mass spectrometer using electron impact ionization. Chemical entities were identified by comparison to metabolomic library entries of purified standards. Following log transformation and imputation with minimum observed values for each compound, data were analyzed using two-way ANOVA with contrasts.

**Statistical analysis**. Statistical analysis was performed using the Prism software (GraphPad, v6, San Diego, USA). Principal component analyses and heatmaps were performed using Qlucore Omics Explorer (Qlucore). Data are plotted in the figures as mean ± s.e.m. Differences between two groups were assessed using Mann–Whitney test. Differences among three or more groups were assessed using one-way ANOVA with Tukey's. Significant differences are indicated in the figures by $*P < 0.05$, $**P < 0.01$, $***P < 0.001$, $****P < 0.0001$. Notable near-significant differences $(0.05 < P < 0.1)$ are indicated in the figures. Notable non-significant (and non-near significant) differences are indicated by "n.s." in the figures.

**Reporting summary**. Further information on research design is available in the Nature Research Reporting Summary linked to this article.

## Data availability

The metabolomics datasets of mouse serum can be accessed at MetaboLights [https://www.ebi.ac.uk/metabolights/index] (Project ID: MTBLS121). The metabolomics datasets of mouse hippocampus tissues can also be accessed at MetaboLights [https://www.ebi.ac.uk/metabolights/index] (Project ID: MTBLS2106). All other data supporting the findings of this study are available from the corresponding author on reasonable request. Source data are provided with this paper.

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

## Acknowledgements

We would like to thank all the members of the Eberl and Lledo laboratories, as well as members of the Peduto laboratory for insightful discussions, and Pr. Peduto for access to the Apotome microscope as well as Giovanni Marsicano and Arnaud Busquets for some advice. A special thanks also to the members of the Pasteur Animal Facility who were essential for this project, and in particular Marion Bérard, Martine Jacob, Thierry Angelique, and Eddie Maranghi. This work was supported by the Major Federating Program "Microbes & Brain" of the Institut Pasteur, Agence Nationale de la Recherche Grant ANR-16-CE15-0021-02-PG-Brain and a grant from the Fédération pour la Recherche sur le Cerveau (FRC). Lledo's laboratory is supported by Agence Nationale de la Recherche Grants ANR-16-CE37-0010-ORUPS and ANR-15-NEUC-0004-02 "Circuit-OPL," Laboratory for Excellence Programme "Revive" Grant ANR-10-LABX-73, and the Life Insurance Company AG2R-La Mondiale. Boneca's laboratory and A.R. were supported by the French Government's Investissement d'Avenir program, Laboratoire d'Excellence "Integrative Biology of Emerging Infectious Diseases" (grant no. ANR-10-LABX-62-IBEID).

## Author contributions

G.C., G.E., and P.-M.L. conceived the study; G.C. established the methodology; G.C., E.S., M.P., L.G.-M., T.L., G.L., A.R., B.C., A.P., E.C.-V., and A.M. performed the experiments; F.L. provided the *L. plantarum*$^{WJL}$ strain; G.C. wrote the original manuscript, which was edited by all authors; G.C., G.E., P.M.L., I.G.B., and G.L. secured funds; G.E., P.-M.L., I.G.B., and C.D. provided resources; G.C., G.E., P.-M.L, G.L., I.G.B., and C.D. supervised the project.

## Competing interests

The authors declare no competing interests.
