## [Peer Review File · Nature Communications]

Reviewers' comments:

Reviewer #1 (Remarks to the Author):

Chevalier and collaborators report evidence suggesting a link between the gut microbiota, the hippocampal endocannabinoid (ECB) system and the development of depressive symptoms in mice subjected to chronic mild stress (CMS). The suggested link is provided by a decrease in availability of the ECB precursor arachidonic acid. Indeed, the main claim of this study is that the behavioral consequences of CMS are "due to lower peripheral levels of fatty acid precursors of ECB ligands" (Summary). If it were fully supported by the data, this conclusion would be of considerable interest. As it stands, however, several key pieces of information are missing, as detailed below.

1. A crucial weakness of the study is that it provides no information on the levels of polyunsaturated fatty acid (PUFA) stores in the hippocampus (or other brain regions potentially involved in the observed effects). In addition to free arachidonic acid -- which was measured in plasma, but not in hippocampal tissue -- the authors should also measure (phospho)lipids that contain arachidonic acid at the sn-2 position and could thus serve as precursors for 2-AG. These include of course phosphatidylinositol-4,5-bisphosphate and 1-stearoyl-2-arachidonoyl-glycerol, but also other phospholipid species that could potentially replenish the arachidonic acid pool via remodelling (e.g., phosphatidylcholine). Of note, even if arachidonic acid levels are abnormally low in CMS mice, other PUFAs could replace it, for example eicosapentaenoic acid, and still give rise to bioactive ECB ligands. So, a more thorough lipidomic analysis must be conducted before firm conclusions can be drawn about the existence of an 'arachidonic acid deficiency syndrome'.

2. Another weakness is that the authors focus solely on 2-AG and do not take into consideration the other main ECB ligand, anandamide, which has been previously shown to be involved in the modulation of depressive symptoms in the CMS model (e.g., Bortolato et al. Antidepressant-like activity of the fatty acid amide hydrolase inhibitor URB597 in a rat model of chronic mild stress. *Biol Psychiatry*. 2007 62:1103-10). This gap is relevant, because it is entirely possible for a deficit in 2-AG signaling to be compensated by an increase in signaling via anandamide.

3. The finding that mTOR signaling is impaired in CMS mice is interesting, but hardly a sure indication that CB1 receptors are hypofunctional. There are many other neurotransmitter and hormone receptors that engage the mTOR pathway.

4. I commend the authors for having tested both globally active and peripherally restricted CB1 receptor antagonists. This said, they would have made a stronger case if they had also tested their hypothesis in CB1 ko mice or other genetic models of 2-AG hypo/hyper-functionality.

5. The arachidonic acid replenishing experiment is unconvincing, for three reasons. First, the authors need to show that replenishing has actually occurred at relevant sites, i.e., plasma and hippocampus. Second, standard chow diets contain fairly high levels of PUFA (from lard or soybean), so it is not clear to me how much additional arachidonate is being provided. Finally, PUFA are absorbed as complex lipids, as they are present in food, not as free acids. Free acids are strongly bioactive and/or can be rapidly converted, in the gut, into a host of bioactive metabolites which could confound interpretation of the results.

A few minor points:

1-AG is not a 'more stable metabolite' of 2-AG but rather a product of chemical isomerization of 2-AG, which is generally formed during sample workup. Since the fraction of 2-AG that undergoes isomerization can vary from one experiment to another, the two isomers must be summed to obtain the correct value of initial 2-AG. Figure 1E is therefore incorrect.

2. 2-OG and 2-PG are not ECBs in that they have direct or indirect interaction with the ECB system

. By the way, I very much doubt the authors' identification of 2-PG. I suspect is 1-PG instead (palmitic acid is most often, if not exclusively, esterified to the sn-1, not the sn-2 position of phospholipids).

Reviewer #2 (Remarks to the Author):

This manuscript presents interesting findings from a well designed preclinical study demonstrating a role of altered endocannabinoid signaling in the development of depression like behavior in mice, which involves a stress induced alteration in gut microbial composition, specifically a reduction of the relative abundance of Lactobacilli. The study confirms several previous observations about the effect of a chronic variable stress paradigm on rodent behavior, gut microbial composition, endocannabinoid system and affective behavior. The major novelty of this study is the data linking the reduction in Lactobacillus relative abundance with a reduction in endocannabinoid precursors and hippocampus endocannabinoid levels. The enthusiasm of this reviewer is somewhat reduced by several concerns:

1. It has previously been demonstrated that the reduction of lactobacillus abundance by chronic variable stress model leads to depression like behavior by a different mechanism, e.g. alteration in the kynurenin pathway. The discussion does not address if reduced relative abundance of the same microbial species could affect the brain by multiple mechanisms.
2. It remains unclear by which specific mechanism the stress induced reduction in Lactobacillus species reduces the lipid precursors of endocannabinoids.
3. When these precursors are administered to the stress animals, they are presumably absorbed in the small intestine. Do the authors assume that the stress effects on the gut microbiota are occurring in the small intestine, reducing absorption of lipids, or in the large intestine which is not the primary site for absorption?
4. What effect could the stress induced change in small intestinal transit have on both lipid absorption?
5. It is surprising that the chronic stress had no effect on intestinal permeability, which is usually observed in other stress models.
6. The 8 week delay in the development of depression like behavior following FMT into the recipient animals is surprising, and should be discussed.
7. Even though several meta analyses suggest that "probiotic" intake has an antidepressant effect, there is a need for caution when taking the reported results as evidence for a meaningful "psychobiotic" effect. None of the quoted studies would measure up to the high quality RCT that have been reported on the effect of pharmacological antidepressants.

Reviewer #3 (Remarks to the Author):

The authors present novel findings usually an established paradigm for depression, UCMS and its transfer through microbiome transfer. The paper is interesting and novel but requires additional data to be suitable for Nature Communication. What happened to the other classes biological systems relevant to depression, HPA axis and inflammation? Were those involved upstream or downstream of the endocannabinoids effects? what is the mechanisms by which UCMS change the microbiome (if this is truly what happens)?

Point-by-point rebuttal

Reviewer #1

1. A crucial weakness of the study is that it provides no information on the levels of polyunsaturated fatty acid (PUFA) stores in the hippocampus (or other brain regions potentially involved in the observed effects). In addition to free arachidonic acid -- which was measured in plasma, but not in hippocampal tissue -- the authors should also measure (phospho)lipids that contain arachidonic acid at the sn-2 position and could thus serve as precursors for 2-AG. These include of course phosphatidylinositol-4,5-bisphosphate and 1-stearoyl-2-arachidonoyl-glycerol, but also other phospholipid species that could potentially replenish the arachidonic acid pool via remodelling (e.g., phosphatidylcholine). Of note, even if arachidonic acid levels are abnormally low in CMS mice, other PUFAs could replace it, for example eicosapentaenoic acid, and still give rise to bioactive ECB ligands. So, a more thorough lipidomic analysis must be conducted before firm conclusions can be drawn about the existence of an 'arachidonic acid deficiency syndrome'.

Reply: Following the referee's suggestion, we have analyzed PUFAs in the hippocampus of mice that have received control microbiota, UCMS microbiota and UCMS microbiota supplemented with AA or *L. plantarum*. We observed a decrease of both n3 and n6 PUFAs in the hippocampus of UCMS microbiota recipient mice when compared to control mice (two-way ANOVA : effect of microbiota, $F(1, 40) = 5.664$; $P = 0.0222$) although this trend was not significant on each PUFA taken individually. When UCMS microbiota recipient mice received supplemented food, we observed a significant normalization of PUFA levels in their hippocampus, both with AA and even more with *L. plantarum* (two-way ANOVA : effect of treatment, $F(3, 80) = 10.90$ $P < 0.0001$). Taken together, we conclude that the general effects of UCMS microbiota on the plasmatic levels of PUFAs is also valid in the hippocampus parenchyma. These results are now mentioned in the revised manuscript (see Suppl. Fig. 6A).

2. Another weakness is that the authors focus solely on 2-AG and do not take into consideration the other main eCB ligand, anandamide, which has been previously shown to be involved in the modulation of depressive symptoms in the CMS model (e.g., Bortolato et al. Antidepressant-like activity of the fatty acid amide hydrolase inhibitor URB597 in a rat model of chronic mild stress. *Biol Psychiatry*. 2007 62:1103-10). This gap is relevant, because it is entirely possible for a deficit in 2-AG signaling to be compensated by an increase in signaling via anandamide.

Reply: We agree with the referee and have performed further experiments to quantify anandamide (AEA) levels in the hippocampus of mice that received control microbiota, UCMS microbiota and UCMS microbiota associated with supplemented with AA or *L. plantarum*. We did not observe a decrease in AEA in the hippocampus of UCMS-microbiota recipient mice compared to control-microbiota recipient mice (see new data presented in Figure 2F). This observation is consistent with previous reports in UCMS mice (Hill et al., 2005, *Neuropsychopharmacology*), in patients suffering from depression (Hill et al., 2008, *Pharmacopsychiatry*) or from healthy subjects exposed to chronic stressors (Yi et al., 2016, *Neuropsychopharmacol. Biol. Psychiatry*). On the other hand, a mere complementation with AA or with *L. plantarum* was sufficient to normalize and even increase the levels of hippocampal AEA (see new data presented in Suppl. Fig. 6B). We conclude that the cellular and behavioral phenotype induced by the UCMS microbiota do not rely on AEA level but rather on 2-AG. This observation is further validated by the fact that JZL184 treatment alone is sufficient to restore a normal phenotype in UCMS-microbiota recipient mice. On the other hand, we discussed the possibility that complementation with AA or *L. plantarum* may have a

broader effect on the CB1 signaling pathway by enhancing the production of both 2-AG and AEA in the brain (see discussion section, page 14).

3. *The finding that mTOR signaling is impaired in CMS mice is interesting, but hardly a sure indication that CB1 receptors are hypofunctional. There are many other neurotransmitter and hormone receptors that engage the mTOR pathway.*

Reply: We have shown that mTOR signaling is impaired in the hippocampus of UCMS mice or mice recipient of UCMS microbiota and we agree that several other receptors might be able to engage the mTOR pathway. However, the fact that enhancing the level of 2-AG (by inhibiting its degradation with JZL184) is necessary to restore hippocampal mTOR signaling while the CB1 antagonist rimonabant is sufficient to decrease mTOR signaling, support the notion that CB1 receptor is a major player to mediate mTOR signaling in our system. Other studies have also observed that CB1 is a major activator of the mTOR pathway in the hippocampus (Busquets-Garcia *et al.*, 2013, *Nat. Med.*). Moreover, it has been shown previously that hippocampus-specific deletion of mTOR abrogates also the effect of JZL184 treatment (Zhong *et al.* 2014, *Neuropsychopharmacology*). Therefore, we believe that the observed alterations in the mTOR pathway are causally linked to CB1 signaling, but the revised manuscript discusses also other alternatives (see discussion page 15).

4. *I commend the authors for having tested both globally active and peripherally restricted CB1 receptor antagonists. This said, they have made a stronger case if they had also tested their hypothesis in CB1 ko mice or other genetic models of 2-AG hypo/hyper-functionality.*

Reply: We agree with the referees that a genetic strategy would bring further confirmation of our results. However such a strategy may not be straightforward and conclusive because of indirect compensatory effects or dose-dependent effects. For instance, the genetic deletion of MAGL, the 2-AG degrading enzyme, enhanced endocannabinoid levels in the brain but also induced CB1R desensitization, which ultimately lead to anxiety and depression (Imperatore *et al.*, 2015, *J Neurochem.*).

In the same vein, high doses of CB1 agonists or cannabinoids (but not low doses) can produce anxiogenic and depression-like effects in mice (Viveros *et al.*, 2005, *Pharmacol. Biochem. Behav.*; Platel & Hillard, 2006, *J. Pharmacol. Exp. Therap.*; Onaivi *et al.*, 1990 *J. Pharmacol. Exp. Therap.*; Moreira *et al.*, 2009, *Best Pract. Res. Clin. Endo. Metab.*). In the same line, high doses of CB1R antagonist AM251 or CB1R full Knock-Out mice display a reduced immobility time compared to littermates in FST (Shearman *et al.*, 2003, *Behav. Pharmacol.*; Tzavara *et al.*, 2003, *Br. J. Pharmacol.*; Griebel *et al.*, 2005, *Biol. Psychiatry*), a result that is opposite to the classically described antidepressant effects of CB1R agonists. Together, these results indicate that the genetic deletion of CB1R may not be the ideal model to test the effect of a CB1 hypo-functionality on depressive-like behaviors and suggest that low CB1 functionality versus CB1 blockade can lead to opposite effects.

To avoid these caveats, we tested whether a more targeted deletion of CB1R in the hippocampus may be better selective strategy compared to the full CB1R-KO. We stereotaxically injected an AAV vector expressing the CRe recombinase under a generic neuronal promoter (AAV9-hSyn-CRe-WPRE, $\sim 10^{11}$ viral genomes/ml) in the dorsal and ventral hippocampus of CB1-fl/fl mice (mouse line from G. Marsicano, Univ. Bordeaux; Marsicano *et al.*, 2003, *Science*) versus control WT littermates. In some animals we also co-injected a CRe-dependant TdTomato-expressing AAV to validate the genetic recombination and its spread (see panel A of the Figure below). Four weeks and 8 weeks post-injection in the hippocampus, we assessed depressive-like behaviors with FST and TST. We observed a reduced immobility time in hippocampal CB1 deleted mice that progressively increase with time, as observed in CB1-KO mice (see panel B in the figure below). Given these results, we concluded that the baseline changes in depressive-like behaviors in CB1-KO, or in hippocampal CB1 deleted mice, preclude any further analysis of the effects of UCMS

microbiota. Nevertheless, we now discuss the dose-dependent effects on CB1-dependent depressive-like symptoms (see discussion section, page 15).

5. The arachidonic acid replenishing experiment is unconvincing, for three reasons. First, the authors need to show that replenishing has actually occurred at relevant sites, i.e., plasma and hippocampus. Second, standard chow diets contain fairly high levels of PUFA (from lard or soybean), so it is not clear to me how much additional arachidonate is being provided. Finally, PUFA are absorbed as complex lipids, as they are present in food, not as free acids. Free acids are strongly bioactive and/or can be rapidly converted, in the gut, into a host of bioactive metabolites which could confound interpretation of the results.

Reply: We agree with the reviewer that food supplementation with AA might indirectly causally link AA deficiency with mood states, for the reasons she/he mentioned. To better characterize the effect of AA supplementation on brain lipids, we performed additional experiments to quantify hippocampal PUFAs in response to the AA treatment. We now report that the hippocampal PUFAs levels in UCMS microbiota recipient mice supplemented with AA, compared to UCMS microbiota recipient mice, are significantly higher (two-way ANOVA : effect of AA treatment compared to UCMS microbiota alone: $F(1, 40) = 10.79, P = 0.0021$). We conclude that AA supplementation in the food is sufficient to normalize hippocampal levels of PUFAs. We now mention these new results in the revised manuscript (see Suppl. Fig. 6A).

A few minor points:

1. 1-AG is not a 'more stable metabolite' of 2-AG but rather a product of chemical

isomerization of 2-AG, which is generally formed during sample workup. Since the fraction of 2-AG that undergoes isomerization can vary from one experiment to another, the two isomers must be summed to obtain the correct value of initial 2-AG. Figure 1E is therefore incorrect.

Reply: Our comments to Fig. 1E refer to a previous publication by Docs *et al.* (*Front. Cell. Neurosci.*, 2017), which states: “2-AG is prone to molecular rearrangement in water-based media, i.e., the arachidonyl moiety moves from the 2-position to the 1-position of glycerol. This non-enzymatic isomerization, known as acyl migration, results in the formation of 1-arachidonoyl-sn-glycerol (1-AG), which is thermodynamically more stable than 2-AG, and the reaction proceeds until it reaches an equilibrium at 1:9 ratio of 2-AG and 1-AG”. Nevertheless, to comply with the referee’s comment, our initial comment “*more stable metabolite*” has been removed from the revised manuscript and replaced by the reviewer’s suggestion “*product of chemical isomerization of 2-AG*” (see page 9).

2. 2-OG and 2-PG are not ECBs in that they have direct or indirect interaction with the ECB system. By the way, I very much doubt the authors' identification of 2-PG. I suspect is 1-PG instead (palmitic acid is most often, if not exclusively, esterified to the sn-1, not the sn-2 position of phospholipids).

Reply: After careful review of the literature, we agree with Reviewer #1 that 2-OG and 2-PG have probably no effect, direct or indirect, on the eCB system (Murataeva *et al.*, 2016, *Pharmacol. Res.*). In accordance with her/him, we have removed this statement from the revised manuscript.

We are grateful from Reviewer #1 for his/her constructive comments on the manuscript.

Reviewer #2

1. It has previously been demonstrated that the reduction of lactobacillus abundance by chronic variable stress model leads to depression like behavior by a different mechanism, e.g. alteration in the kynurenine pathway. The discussion does not address if reduced relative abundance of the same microbial species could affect the brain by multiple mechanisms

Reply: We agree that other metabolic pathways might also support our behavioral observations. For this reason, we have further investigated the kynurenine pathway and monitor the plasma levels of kynurenine and did not report significant changes in mice groups (new data presented in Suppl. Fig. 3L). This result, together with the fact that AA food supplementation is sufficient to complement the UCMS microbiota, indicate that the perturbation of lipid metabolism is sufficient to account for the observed phenotype in UCMS mice, without ruling out other alternative pathways (see discussion section, page 17).

2. It remains unclear by which specific mechanism the stress induced reduction in *Lactobacillus* species reduces the lipid precursors of endocannabinoids.

Reply: The mechanisms by which stress induces a reduction in *Lactobacilli* and lipid precursors are still unclear and may involve alterations of the immune system, the digestive system, and/or hormonal levels. This question is beyond the scope of this study which is centered on the effects of UCMS on brain and behavior. However, potential mechanisms are further discussed in the revised manuscript. In particular, we comment previous results illustrating how *Lactobacilli* modulate host lipid metabolism (e.g., Chiu *et al.*, 2006, *Appl. Microbiol. Biotechnol.*; Kishino *et al.*, 2013, *Proc. Natl. Acad. Sci.* and in particular L.

plantarum in Bao *et al.*, 2012, *Eur. J. Lipid Sci. Technol.*; Xie *et al.*, 2011, *BMC Complement. Altern. Med.*; see page 18).

3. When these precursors are administered to the stress animals, they are presumably absorbed in the small intestine. Do the authors assume that the stress effects on the gut microbiota are occurring in the small intestine, reducing absorption of lipids, or in the large intestine which is not the primary site for absorption?

Reply: As highlighted by the reviewer, *Lactobacilli* are more prominent in the small intestine (SI), as compared to other phyla in the small intestine (SI). We assume that the differences observed in the feces mirror the decrease of *Lactobacilli* in the SI, and therefore lipid absorption (see for instance El Aidy *et al.*, 2015, *Curr Opin. Biotechnol.*). This point is further discussed in the revised manuscript (see page 18).

4. What effect could the stress induced change in small intestinal transit have on both lipid absorption?

Reply: A previous study revealed that chronic intermittent stress in mice increases fecal excretion of bile acids, thus altering cholesterol pathway in the intestine (Silvennoinen *et al.*, 2015, *Physiol Rep.*). In humans, measuring intestinal transit in affective disorders has demonstrated that mood has an effect on intestinal motor function (e.g., Gorard *et al.*, 1996, *Gut*). Because lipid absorption depends on intestinal transit time, we now mention these studies as a potential mechanism (see discussion section, page 17-18). Additional mechanisms include a decline in food-motivated operant behavior in depressive-like mice and disruption of the regulation of lipid synthesis as reported during a chronic stress (Chuang, *et al.*, 2010, *J. Lipid Res.*). These possibilities are now mentioned in the revised manuscript (see discussion section page 18). Collectively, our study supports previous investigations showing that specific lipid classes are directly involved in depression and anxiety disorders, paving the way for new lipid-based targets for mood disorder prevention and/or treatment (for a review, see Müller *et al.*, 2015, *BBA - Molecular and Cell Biology of Lipids*).

5. It is surprising that the chronic stress had no effect on intestinal permeability, which is usually observed in other stress models.

Reply: The unpredictable chronic mild stress (UCMS) is one out of several animal models of depression. It is based on chronic exposure to unpredictable stressors. Alternative models use different stress procedures with stronger severity. The fact that UCMS provides distinct physiological changes compared to other stressful approaches has been a matter of debate (see for a recent review Willner, 2017, *Neurobiol Stress.*). We now provide the reader with some reviews that support the fact that changes induced by “mild” long-lasting stressors differ from those induced by more acute and intense stressors (Antoniuk *et al.*, 2019, *Neurosci Biobehav Rev.*; Willner, 2017, *Neurobiol Stress.*; see discussion section page 15).

6. The 8-week delay in the development of depression-like behavior following FMT into the recipient animals is surprising, and should be discussed.

Reply: We point the reviewer to the fact that the mouse strain used in this study are C57Bl/6, which display minimal behavioral changes in response to environmental stressors and are considered relatively stress-resistant (compared to swiss mice or BalB/C mice : Anisman *et al.*, 1998, *Stress*; Shanks *et al.*, 1990, *Pharmacol. Biochem. Behav.*), likely due to a blunted corticosterone response (Flint & Tinkle, 2001, *Toxicol. Sci.*). In that line, Monteiro *et al.* have shown that C57Bl/6 mice are quite resistant to the commonly used 4-

week-long UCMS protocol and require a 8-week-long protocol to induce the canonical chronic-stress associated responses (Monteiro *et al.*, 2015, *Front. Psychiatry*). In addition, considering the technical constraint of keeping mice in isolators to control their microbial environment, we wait until 8 weeks to stabilize the associated homeostatic processes before manipulating them for behavioral assays. Therefore, our study does not suggest that the development of depression-like behavior requests 8 weeks but this delay was rather imposed by our methodology. We have clarified this issue in the revised version (see Materials & Methods section, page 27).

7. Even though several meta analyses suggest that “probiotic” intake has an antidepressant effect, there is a need for caution when taking the reported results as evidence for a meaningful “psychobiotic” effect. None of the quoted studies would measure up to the high quality RCT that have been reported on the effect of pharmacological antidepressants.

Reply: We agree with the reviewer on the need to be cautious about “psychobiotic” effect and therefore emphasize this point in the discussion (see discussion section, page 17). We are grateful from Reviewer #2 for his/her constructive comments on the manuscript.

Reviewer #3

1. What happened to the other classes biological systems relevant to depression, HPA axis and inflammation?

Reply: To address this question, we looked at the corticosterone levels in recipient mice and did not observe significant differences in the serum, suggesting that the HPA axis was not chronically overactivated in those mice even though we cannot rule out that early differences may have taken place shortly after FMT (new data presented in Suppl. Fig. 3M). Similarly, we analyzed the composition of the adaptive immune system in the small intestine of recipient mice and did not observe any significant differences regarding T cells (Suppl. Fig. 3A), B cells (Suppl. Fig. 3B), CD4 T cells (Suppl. Fig. 3C), CD8 T cells (Suppl. Fig. 3D) or Treg (Suppl. Fig. 3E). However, to make sure that the immune system was not skewed toward a certain type of immunity, we analyzed more in depth innate and adaptive immune system in the gut of recipient mice, especially regarding a potential skewing toward type-1, type-2 or type-3 immunity. Once again, we did not observe any significant differences regarding Th1 cells (Suppl. Fig. 3F), innate lymphoid cells type 1 (ILC1) (Suppl. Fig. 3I), Th2 cells (Suppl. Fig. 3G), ILC2 (Suppl. Fig. 3J), Th17 cells (Suppl. Fig. 3H) and ILC3 (Suppl. Fig. 3K). Therefore, we conclude that the immune system is not involved in the behavioral differences observed between control microbiota and UCMS microbiota recipient mice, at least after 8 weeks. These new results have been added in the revised version (see Suppl. Fig. 3A-K).

2. Were those involved upstream or downstream of the endocannabinoid effects?

Reply: Regarding the results presented above, we assume that neither HPA nor the immune system are involved in our observations, at least at the time point studied (*i.e.*, 8 weeks post-FMT).

3. What is the mechanisms by which UCMS change the microbiome (if this truly what happens)?

Reply: The mechanism by which stress induces intestine dysbiosis has not yet been figured out. It may involve subtle changes of the gut homeostasis, alterations of the enteric nervous

and immune systems, or changes in some metabolic pathways. This is now further commented in the discussion part (see discussion section page 17-18). We are grateful from Reviewer #3 for his/her constructive comments on the manuscript.

Reviewers' comments:

Reviewer #1 (Remarks to the Author):

The authors have only partially addressed my concerns.

1. The lipid analyses remain restricted to free (non-esterified) PUFA. As pointed out in my critique, free PUFA represent a minor fraction of the total PUFA pool, the vast majority of which is found in complex lipids such as phospholipids. Without a quantitative analysis of the complex lipid pool the lipidomics data presented are virtually meaningless and the claim of 'an arachidonic acid deficiency syndrome' unwarranted.

2. In response to my request, the authors have measured anandamide levels in the hippocampus. They show that anandamide levels are not changed in the hippocampus of UCMS-microbiota recipient mice. Based on this finding, how can they claim that AA or L plantarum supplementation 'normalized' anandamide levels? It seems to me that L plantarum supplementation simply increased such levels.

3. The authors have satisfactorily addressed this point.

4. It is unfortunate that genetic models cannot be brought to bear to strengthen the authors' conclusions.

5. I find it extremely surprising that adding a relatively small amount of AA to food can, as the authors claim, "normalize hippocampal levels of PUFAs". As pointed out before, this claim is groundless if lipidomics analyses are limited to free PUFA.

Minor point 1. My main point, namely that quantifying 2-AG requires summing 2-AG and 1(3)-AG was not addressed. Please note that I consider this a 'minor point' only in the sense that is easily addressed. A quantification of 2-AG that does not include 1(3)-AG is incorrect.

Minor point 2. I am glad that the authors performed a 'careful review of the literature' to verify that my comment about 2-OG was correct. Still, they decided to ignore my (more important) comment about '2-PG'. I don't exclude the existence of such a MAG, I just find it very implausible. The authors need either to verify it by comparison with an authentic standard, or simply drop the 2- and refer to the species as PG.

Reviewer #2 (Remarks to the Author):

The authors have satisfactorily responded to all my comments. The only remaining comment I have is related to the way the authors refer to the clinical effectiveness of probiotics in the treatment of depression. Even though there are now several publications of varying quality showing a beneficial effect of some probiotics on symptoms of depression, I do not believe that the effect is clinically meaningful to include probiotics in clinical treatment guidelines for depression or anxiety

Reviewer #3 (Remarks to the Author):

I am satisfied with the revision

Reviewer #1 (Remarks to the Author):

The authors have only partially addressed my concerns.

1. The lipid analyses remain restricted to free (non-esterified) PUFA. As pointed out in my critique, free PUFA represent a minor fraction of the total PUFA pool, the vast majority of which is found in complex lipids such as phospholipids. Without a quantitative analysis of the complex lipid pool the lipidomics data presented are virtually meaningless and the claim of 'an arachidonic acid deficiency syndrome' unwarranted;

Reply: We would like to reassure reviewer #1 that there is no experimental bias in our lipid analyses since PUFA's were obtained after hydrolysis of all extracted lipid molecular species (including glycerolipids and glycerophospholipids) from mice brain hippocampus in order to obtain a both free and esterified PUFA pool. This is mentioned in the material and method section (page 42, lines 984-988 of the present manuscript), in the result section (page 12 line 256 and page 13, line 290) and the legend of Suppl. Fig. 4E and 6A.

2. In response to my request, the authors have measured anandamide levels in the hippocampus. They show that anandamide levels are not changed in the hippocampus of UCMS-microbiota recipient mice. Based on this finding, how can they claim that AA or L plantarum supplementation 'normalized' anandamide levels? It seems to me that L plantarum supplementation simply increased such levels.

Reply: Following the referee's comment, we have removed the term "normalisation" from the text and simply refer to an "increase" of anandamide levels (see result section page 12, line 256-261 ; page 13, line 290-294). This increase in anandamide is also discussed in the discussion section (page 14, line 318-320).

3. The authors have satisfactorily addressed this point.

4. It is unfortunate that genetic models cannot be brought to bear to strengthen the authors' conclusions.

5. I find it extremely surprising that adding a relatively small amount of AA to food can, as the authors claim, "normalize hippocampal levels of PUFAs". As pointed out before, this claim is groundless if lipidomics analyses are limited to free PUFA.

Reply: This point regarding free PUFA has been addressed above (see point#1). We have also removed the term "normalisation" and simply refer to an "increase" of PUFA.

Minor point 1. My main point, namely that quantifying 2-AG requires summing 2-AG and 1(3)-AG was not addressed. Please note that I consider this a 'minor point' only in the sense that is easily addressed. A quantification of 2-AG that does not include 1(3)-AG is incorrect.

Reply: Given the potential isomerization between 1(3)-AG and 2-AG following acetonitrile

precipitation, the content of both 2-AG and 1(3)-AG isomers were summed for quantification. This is now mentioned in the material and methods section (page 41, line 979-981) and in the results section (page 9, line 188-189).

Minor point 2. I am glad that the authors performed a 'careful review of the literature' to verify that my comment about 2-OG was correct. Still, they decided to ignore my (more important) comment about '2-PG'. I don't exclude the existence of such a MAG, I just find it very implausible. The authors need either to verify it by comparison with an authentic standard, or simply drop the 2- and refer to the species as PG.

Reply: 2-OG and 2-PG cannot be considered as endocannabinoids (they do not bind to the endocannabinoid receptors) but rather as "congeners" of 2-AG (Murataeva et al., Pharmacol Res. 2016 110:173-180). For sake of clarification, we have decided to remove the quantification of 2-OG and 2-PG in the main figure and focus on 2-AG and AEA, the two main endocannabinoids.

Reviewer #2 (Remarks to the Author):

The authors have satisfactorily responded to all my comments. The only remaining comment I have is related to the way the authors refer to the clinical effectiveness of probiotics in the treatment of depression. Even though there are now several publications of varying quality showing a beneficial effect of some probiotics on symptoms of depression, I do not believe that the effect is clinically meaningful to include probiotics in clinical treatment guidelines for depression or anxiety.

Reply: In the revised version, we tune down our initial statement and mentioned that the efficacy of probiotics compared to antidepressants is a matter of debate (see discussion section, page 17, line 372-373: "dietary or probiotic interventions might be complement tools to increase efficiency of classical therapeutical approaches" page 18, line 409-410).

REVIEWER COMMENTS

Reviewer #1 (Remarks to the Author):

The authors have addressed all my concerns.

Reviewer #4 (Remarks to the Author):

We were asked to comment specifically on the technical aspects of the microbiota component of the manuscript. Addressing the following controls and methodological details would increase the rigor of the study.

Can the authors please provide data for profiling donor and recipient microbiota to confirm high fidelity microbiota transplantation?

In the text, the authors describe that the family Lactobacillaceae reduces with unpredictable chronic mild stress, and later, the authors describe the selection of *Lactobacillus plantarum* based on existing literature on its effects on host physiology. Can the authors comment on whether the specific Lactobacillaceae OTUs identified exhibited homology to *Lactobacillus plantarum* in particular, or disclose whether other Lactobacilli species are also likely candidates?

For the fecal microbiota transplant experiments, could the authors please cite a paper using the same methodology as this paper or state how they tested for microbiota depletion following antibiotic administration? Additionally, could the authors please specify in the methods how the administration was performed (I.e. what mass of donor material was used from each mouse, what solution was used to suspend stool mixed for gavage, etc.)?

In the methods section for 16S rRNA gene sequencing and analysis, could the authors detail whether the n of donors and recipients sequenced was performed on a per cage basis or per mouse within a single cage?

Could the authors please specify in the section on statistical analysis of microbiome composition changes what is meant by "Further statistical analysis was conducted using Prism software"? If space permits, it would be useful to detail what exactly was performed in Prism and whether the taxonomic statistics included correction for multiple comparisons.

Point by point reply to reviewer #4

Reviewer 4

1. Can the authors please provide data for profiling donor and recipient microbiota to confirm high fidelity microbiota transplantation?

To answer the first point of reviewer #4, donor mice microbiota was analyzed prior to transfer to recipient mice. Recipient mice were kept in isolators until behavioral testing and their microbiota analyzed at this time point, 8 weeks after FMT. Therefore, recipient mice from control and UCMS mice microbiota were kept isolated in the same conditions until microbiota analysis. When comparing the abundance of different families between donor and recipient mice for control microbiota and UCMS microbiota FMT, we did not observe significant differences between donor and recipient mice (figures below, same data than Fig5g):

However, we acknowledge reviewer #4 for his/her comment but considering the present difficulties due the COVID-19 crisis, we are afraid we cannot perform 16S analysis from feces of recipient mice at earlier time point than the one provided.

2. In the text, the authors describe that the family *Lactobacillaceae* reduces with unpredictable chronic mild stress, and later, the authors describe the selection of *Lactobacillus plantarum* based on existing literature on its effects on host physiology. Can the authors comment on whether the specific *Lactobacillaceae* OTUs identified exhibited homology to *Lactobacillus plantarum* in particular, or disclose whether other *Lactobacilli* species are also likely candidates?

We would like to thank reviewer #4 for his/her comment. Taxonomic assignment based on amplification of V3-V4 16S region remains elusive at a lower level than family level. Therefore, we chose to get taxonomy at family level to keep a high level of confidence over identification. As described in the manuscript, we chose *L. plantarum* based on existing literature suggesting a role in regulating fatty acid metabolism and modulating host's lipid composition. However, we fully agree that other *Lactobacilli* species are also candidates to impact positively the host's lipid metabolism and therefore, may be as effective as *L. plantarum* in normalizing the phenotype observed. This is now mentioned in the discussion part of the manuscript (lines 403-406).

3. For the fecal microbiota transplant experiments, could the authors please cite a paper using the same methodology as this paper or state how they tested for microbiota depletion following antibiotic administration?

To answer reviewer #4 point, we refer to the paper by the group of Karine Clement (Le Roy *et al.*, 2018, Comparative Evaluation of Microbiota Engraftment Following Fecal Microbiota Transfer in Mice Models: Age, Kinetic and Microbial Status Matter) in the manuscript (line 780). We used colistin instead of neomycin in the cocktail of antibiotics but used otherwise the same mix.

4. *Additionally, could the authors please specify in the methods how the administration was performed (i.e. what mass of donor material was used from each mouse, what solution was used to suspend stool mixed for gavage, etc.)?*

To answer reviewer #4 point, here is the description added in the M&M part of the manuscript of the method of administration for the FMT (lines 785-790):

Fresh fecal pellets were collected directly from the rectum of donor mice and 100mg (about 5-6 fecal pellets) were homogenized in 1mL sterile saline. Homogenates were then passed through a 20µm pore nylon filter to remove large particulate and fibrous matter. The solution was collected and 200µL was administered by oral gavage to recipient mice, within 15 min to minimize changes in microbial contents (Chang, C. J. et al., 2015, Ganoderma lucidum reduces obesity in mice by modulating the composition of the gut microbiota).

5. *In the methods section for 16S rRNA gene sequencing and analysis, could the authors detail whether the n of donors and recipients sequenced was performed on a per cage basis or per mouse within a single cage?*

We acknowledge reviewer #4 for his/her comment. We performed 16S rRNA sequencing from of donor and recipient mice raised in 2 different cages per group. The data are representative of 2 different experiments, therefore 4 different cage per group.

6. *Could the authors please specify in the section on statistical analysis of microbiome composition changes what is meant by "Further statistical analysis was conducted using Prism software"? If space permits, it would be useful to detail what exactly was performed in Prism and whether the taxonomic statistics included correction for multiple comparisons.*

To answer reviewer #4 comment, we added in the manuscript the following sentence in the "16S Data Analysis" paragraph of the M&M (lines 860-861):

Differences between two groups were assessed using Mann-Whitney test for family abundance.

REVIEWERS' COMMENTS:

Reviewer #4 (Remarks to the Author):

The authors have sufficiently addressed all comments.